# Risk factors for extended-spectrum beta-lactamase (ESBL)-producing *E. coli* carriage among children in a food animal-producing region of Ecuador: A repeated measures observational study

Heather K. Amato[1]*, Fernanda Loayza[2], Liseth Salinas[2], Diana Paredes[2], Daniela Garcia[2], Soledad Sarzosa[2], Carlos Saraiva-Garcia[2], Timothy J. Johnson[3,4], Amy J. Pickering[5,6], Lee W. Riley[7†], Gabriel Trueba[2], Jay P. Graham[1]*

1 Environmental Health Sciences Division, School of Public Health, University of California, Berkeley, California, United States of America, 2 Instituto de Microbiología, Colegio de Ciencias Biológicas y Ambientales, Universidad San Francisco de Quito, Quito, Ecuador, 3 Department of Veterinary and Biomedical Sciences, University of Minnesota, Saint Paul, Minnesota, United States of America, 4 Mid Central Research & Outreach Center, Willmar, Minnesota, United States of America, 5 Department of Civil and Environmental Engineering, University of California, Berkeley, California, United States of America, 6 Blum Center for Developing Economies, University of California, Berkeley, California, United States of America, 7 Division of Infectious Diseases and Vaccinology, School of Public Health, University of California, Berkeley, California, United States of America

† Deceased.

* heather_amato@berkeley.edu (HKA); jay.graham@berkeley.edu (JPG)

**Data Availability Statement:** Raw reads from isolates sequenced in this study are available at the NCBI Short Read Archive (SRA) under BioProject accession no. PRJNA861272. Exposure and outcome data used in epidemiological analyses are

## Abstract

### Background

The spread of antibiotic-resistant bacteria may be driven by human–animal–environment interactions, especially in regions with limited restrictions on antibiotic use, widespread food animal production, and free-roaming domestic animals. In this study, we aimed to identify risk factors related to commercial food animal production, small-scale or "backyard" food animal production, domestic animal ownership, and practices related to animal handling, waste disposal, and antibiotic use in Ecuadorian communities.

### Methods and findings

We conducted a repeated measures study from 2018 to 2021 in 7 semirural parishes of Quito, Ecuador to identify determinants of third-generation cephalosporin-resistant *E. coli* (3GCR-EC) and extended-spectrum beta-lactamase *E. coli* (ESBL-EC) in children. We collected 1,699 fecal samples from 600 children and 1,871 domestic animal fecal samples from 376 of the same households at up to 5 time points per household over the 3-year study period. We used multivariable log-binomial regression models to estimate relative risks (RR) of 3GCR-EC and ESBL-EC carriage, adjusting for child sex and age, caregiver education, household wealth, and recent child antibiotic use. Risk factors for 3GCR-EC included living within 5 km of more than 5 commercial food animal operations (RR: 1.26; 95%

published and publicly available on Dryad (https://doi.org/10.5061/dryad.41ns1rnm7).

**Funding:** This research was supported by the National Institute of Allergy and Infectious Diseases of the National Institutes of Health (NIH) under Award Number R01AI135118, awarded to JG. The funders had no role in study design, data collection and analysis, decision to publish, or preparation of the manuscript.

**Competing interests:** The authors have declared that no competing interests exist.

**Abbreviations:** 3GCR-EC, third-generation cephalosporin-resistant *E. coli*; ARG, antibiotic resistance gene; CFO, commercial food animal operation; CI, confidence interval; CLSI, Clinical and Laboratory Standards Institute; ESBL, extended-spectrum beta-lactamase; ESBL-E, extended-spectrum beta-lactamase producing Enterobacterales; ESBL-EC, extended-spectrum beta-lactamase *E. coli*; ExPEC, extraintestinal pathogenic *E. coli*; LMIC, low- and middle-income country; MLST, multilocus sequence typing; NIH, National Institutes of Health; ODK, Open Data Kit; RERI, relative excess risk due to interaction; RR, relative risk; SARS-CoV-2, Severe Acute Respiratory Syndrome Coronavirus 2; SD, standard deviation; ST, sequence type.

confidence interval (CI): 1.10, 1.45; *p*-value: 0.001), household pig ownership (RR: 1.23; 95% CI: 1.02, 1.48; *p*-value: 0.030) and child pet contact (RR: 1.23; 95% CI: 1.09, 1.39; *p*-value: 0.001). Risk factors for ESBL-EC were dog ownership (RR: 1.35; 95% CI: 1.00, 1.83; *p*-value: 0.053), child pet contact (RR: 1.54; 95% CI: 1.10, 2.16; *p*-value: 0.012), and placing animal feces on household land/crops (RR: 1.63; 95% CI: 1.09, 2.46; *p*-value: 0.019). The primary limitations of this study are the use of proxy and self-reported exposure measures and the use of a single beta-lactamase drug (ceftazidime with clavulanic acid) in combination disk diffusion tests for ESBL confirmation, potentially underestimating phenotypic ESBL production among cephalosporin-resistant *E. coli* isolates. To improve ESBL determination, it is recommended to use 2 combination disk diffusion tests (ceftazidime with clavulanic acid and cefotaxime with clavulanic acid) for ESBL confirmatory testing. Future studies should also characterize transmission pathways by assessing antibiotic resistance in commercial food animals and environmental reservoirs.

## Conclusions

In this study, we observed an increase in enteric colonization of antibiotic-resistant bacteria among children with exposures to domestic animals and their waste in the household environment and children living in areas with a higher density of commercial food animal production operations.

## Author summary

### Why was this study done?

- An estimated 1.27 million deaths in 2019 were attributable to bacterial antibiotic-resistant infections, 89% of which occurred in low- and middle-income countries (LMICs).

- Small-scale and commercial-scale food animal production is expanding rapidly in middle-income countries like Ecuador and antibiotic use in food animals is increasing at the fastest rate in these settings, fueling the emergence and selection of multidrug-resistant bacteria.

- Evidence is needed to inform action against the spread of antibiotic resistance in communities with varying degrees of both small-scale or "backyard" and commercial food animal production, especially in South America where research on community-acquired antibiotic resistance is lacking.

### What did the researchers do and find?

- We conducted a repeated measures study in a food animal-producing region in Ecuador to compare the risk of antibiotic-resistant bacterial (ARB) carriage among children with varying degrees of exposure to backyard and commercial food animals.

- We used multivariable log-binomial regression models to estimate relative risks (RR) of ARB carriage, adjusting for child sex and age, caregiver education, household wealth, and recent child antibiotic use.

- Living within 5 km of >5 commercial food animal operations, household pig or dog ownership, child contact with pets, and placing animal feces on household crops were identified as significant risk factors for ARB carriage in children.

## What do these findings mean?

- This study underscores the need for improved waste management and surveillance of antibiotic resistance in LMICs with widespread food animal production.

- A significant challenge remains: There is a lack of available data on antibiotic usage and resistance in commercial food animal operations and their effluent, in Ecuador and globally, due to limited oversight and surveillance.

- In future studies, researchers should work with local and national governments to monitor antibiotic use and resistance in food animals, food animal production waste, nearby environmental reservoirs, and food animal products in order to more accurately characterize sources of exposure and transmission routes of community-acquired ARB.

- The primary limitations of this study are the use of proxy and self-reported exposure measures and the use of a single beta-lactamase drug (ceftazidime with clavulanic acid) in combination disk diffusion tests for ESBL confirmation, potentially underestimating phenotypic ESBL production among cephalosporin-resistant *E. coli* isolates.

## Introduction

Environmental fecal contamination from food animal production is increasingly recognized as an important contributor to the global antibiotic resistance crisis. Globally, large quantities of clinically important antibiotics are administered to food animals (poultry and livestock raised for meat and dairy products) to promote growth and prevent infection [1]. An estimated 1.27 million deaths in 2019 were attributable to bacterial antibiotic-resistant infections, 89% of which occurred in low- and middle-income countries (LMICs) [2]. Antibiotic-resistant bacteria, including extended-spectrum beta-lactamase producing Enterobacterales (ESBL-E), found in humans have been linked to food animals [3,4]. ESBL-E—deemed a serious threat to global public health by the World Health Organization and the US Centers for Disease Control and Prevention—confer resistance to a broad spectrum of beta-lactam antibiotics including penicillins and cephalosporins, the most commonly used treatments for bacterial infections [5,6].

With the rapid expansion of food animal production in LMICs, antibiotic use in these settings is projected to increase by upwards of 200% from 2010 to 2030, fueling the emergence and selection of these multidrug-resistant bacteria [1]. Estimating the risks of ESBL-E colonization in LMIC communities with exposures to food animal production is a crucial step towards developing strategies to combat the global spread of antibiotic-resistant bacteria. With limited treatment options, cephalosporin-resistant and ESBL-producing bacterial infections result in longer and more costly hospital stays, increased severity of illness, and increased risk of mortality [7–10]. Asymptomatic carriage of ESBL-E in commensal gut bacteria may still

pose a threat to health; ESBLs are frequently encoded by plasmids that facilitate horizontal transfer of resistance genes, allowing commensal bacteria to share ESBL-encoded genes with pathogenic bacteria [11–14]. Horizontal gene transfer rapidly propagates phenotypic resistance among diverse bacteria in animals, the environment, and humans [11,15]. Globally, the prevalence of ESBL-E is increasing; as of 2018, an estimated 20% of healthy individuals harbor ESBL-producing *Escherichia coli* in their guts [16].

In upper middle-income countries like Ecuador, both commercial and small-scale or "backyard" food animal production are increasing as population growth and increasing wealth drive consumption of animal products [17]. Antibiotics are largely unregulated in Ecuador and other LMICs; medically important antibiotics are routinely used in both large-scale and small-scale food animal operations at subtherapeutic doses for growth promotion and disease prevention [18,19]. In Ecuador, an estimated 84% of rural households and 29% of urban households own livestock, and small-scale poultry farmers have reported regularly administering a range of 6 different classes of antibiotics [20,21]. Contact with animal waste is elevated in LMICs, increasing the potential for exposure to ESBL-E and other antibiotic-resistant bacteria. Domestic animals and backyard food animals commonly defecate in the household environment [22,23], and young children are frequently exposed to high doses of poultry, livestock, and domestic animal feces through the consumption of soil and hand-to-mouth behaviors [24,25]. Domestic animals and fecal contamination of household soil, food, and drinking water are common sources of resistant bacteria in LMICs [26–29]. Exposure to animal feces can also increase the risk of diarrhea [30–32]. Epidemiological studies in LMICs have focused on exposures to feces or fecal pathogens, broadly; few studies have assessed exposures and risk factors for antibiotic-resistant and ESBL-E infections among children in LMICs [33–36].

Given the anticipated growth in unrestricted antibiotic use for food animals in LMICs, there is an urgent need to quantify the risks of antibiotic-resistant and ESBL-E carriage and infections among children living near small-scale and/or commercial food animal production in these settings. This study aimed to estimate the risk of cephalosporin-resistant and ESBL-producing *E. coli* carriage among children with varying degrees of exposure to small-scale and/or commercial food animal production in semirural parishes of Quito, Ecuador. We hypothesized a priori that (1) household-level exposures to small-scale food animal production are associated with an increased risk of third-generation cephalosporin-resistant *E. coli* (3GCR-EC) and ESBL-producing *E. coli* (ESBL-EC); (2) exposures to commercial food animal production operations in the community are associated with an increased risk of 3GCR-EC and ESBL-EC; and (3) combined exposures to both small-scale and commercial food animal production are associated with a greater increased risk of 3GCR-EC and ESBL-EC in children than small-scale or commercial food animal exposures, alone.

## Methods

### Ethics statement

This study was approved by the Office for Protection of Human Subjects at the University of California, Berkeley (IRB# 2019-02-11803), the Bioethics Committee at the Universidad San Francisco de Quito (#2017-178M), and the Ecuadorian Health Ministry (#MSPCURI000243-3). Formal written consent was obtained from each primary caregiver of children enrolled in the study prior to participation.

### Study site

This study was carried out in semirural communities east of Quito, Ecuador by researchers at the Instituto de Microbiología at the Universidad San Francisco de Quito (USFQ) and the

University of California, Berkeley School of Public Health. The study area was approximately 320 km$^2$ and included commercial food animal operations and small-scale or "backyard" production of food animals for both subsistence farming and trade. Irrigation canals fed by nearby rivers flow throughout communities in the study area and are used for subsistence farming and small-scale agricultural production.

## Study design

This repeated measures observational study aimed to enroll 360 households through stratified random sampling across 7 semirural parishes. For power calculations and sample size determination, please see SI (Sample Size and Power Calculations in S1 Files). Random sampling was stratified at the neighborhood level (within parishes) based on the following strata, as determined by community-based fieldworkers: (1) backyard food animal production present in neighborhood; (2) backyard food animal production present and commercial food animal production within 1 km of neighborhood; and (3) no backyard food animal production and no commercial production within 1 km. Households were enrolled if they met the following inclusion criteria at the time of enrollment: (1) a primary child caretaker who was over 18 years of age was present; and (2) a child between the ages of 6 months and 5 years old was present in the household. Informed written consent was provided by the child's caretaker prior to participation in the study. If there was more than 1 child at a given household, the youngest child was selected for participation. Due to high rates of migration out of the study site and loss to follow-up after the onset of the Severe Acute Respiratory Syndrome Coronavirus 2 (SARS-CoV-2) pandemic, we enrolled new households each cycle. Households were visited up to 5 times between August 2018 and August 2021 by trained field staff who conducted household surveys and collected household GPS coordinates and biological samples at each visit. This study is reported as per the Strengthening the Reporting of Observational Studies in Epidemiology (STROBE) guideline (Supporting information (SI), Checklist in S1 Files). Analysis plans for this manuscript were developed beginning in May 2020 and were last revised in March 2021 after sequencing results became available; we amended our analysis plan to include descriptive results of antibiotic resistance genes (ARGs) identified from sequenced *E. coli* isolates. This research was supported by the National Institute of Allergy and Infectious Diseases of the National Institutes of Health (NIH) under Award Number R01AI135118.

## Exposure assessment

The primary exposures of interest were exposures to commercial food animal production in the community and household-level exposures to small-scale food animal production. Exposures and other household characteristics and practices were assessed at each household visit to capture time-varying exposures. Exposure to commercial food animal production was assessed in 3 ways: (1) distance to the nearest commercial food animal production facility; (2) density of commercial food animal production facilities; and (3) proximity to drainage paths of commercial food animal production facilities. Commercial poultry production facilities—vertically integrated operations marked by long barns with a metal roof that typically held approximately 20,000 birds or more—were located using satellite imagery in Google Maps and confirmed as active operations through site visits and ground-truthing. Other types of food animal production operations were identified through local knowledge and ground-truthing.

For the first measure of commercial food animal production exposure, Euclidean distance between each household and the nearest active commercial food animal operation was measured at each time point. For the second measure of exposure, the density of commercial food animal operations was assessed by summing the number of operations within a five-kilometer

buffer of each household at each time point. We used the *sf* package in R to create these first 2 exposure variables [37,38]. While proximity may serve as a proxy for the likelihood of environmental contamination from nearby commercial operations, density may serve as a proxy for the extent of environmental contamination from nearby operations. Density of commercial poultry operations has been associated with cephalosporin-resistant *E. coli* in nearby stream water and sediment in the United States [39]. Finally, for the third exposure measure, drainage paths from each commercial food animal operation were identified using ArcGIS Online Trace Downstream tool, which uses a digital elevation model to identify downstream flow paths from elevation surfaces, drainage directions, river networks, and watershed boundaries obtained from the HydroSHEDS 90 m database [40]. We then created buffers around drainage paths, which ranged from 2.9 to 3.1 km in length, and spatially joined study households to buffer layers in QGIS to identify which households were within 100 or 500 meters of commercial operation drainage paths [41] (Fig 1). Proximity to these drainage paths may capture exposures to antibiotic-resistant bacteria transported from commercial food animal operations through waterways, even when households are located further from the food animal operation, itself.

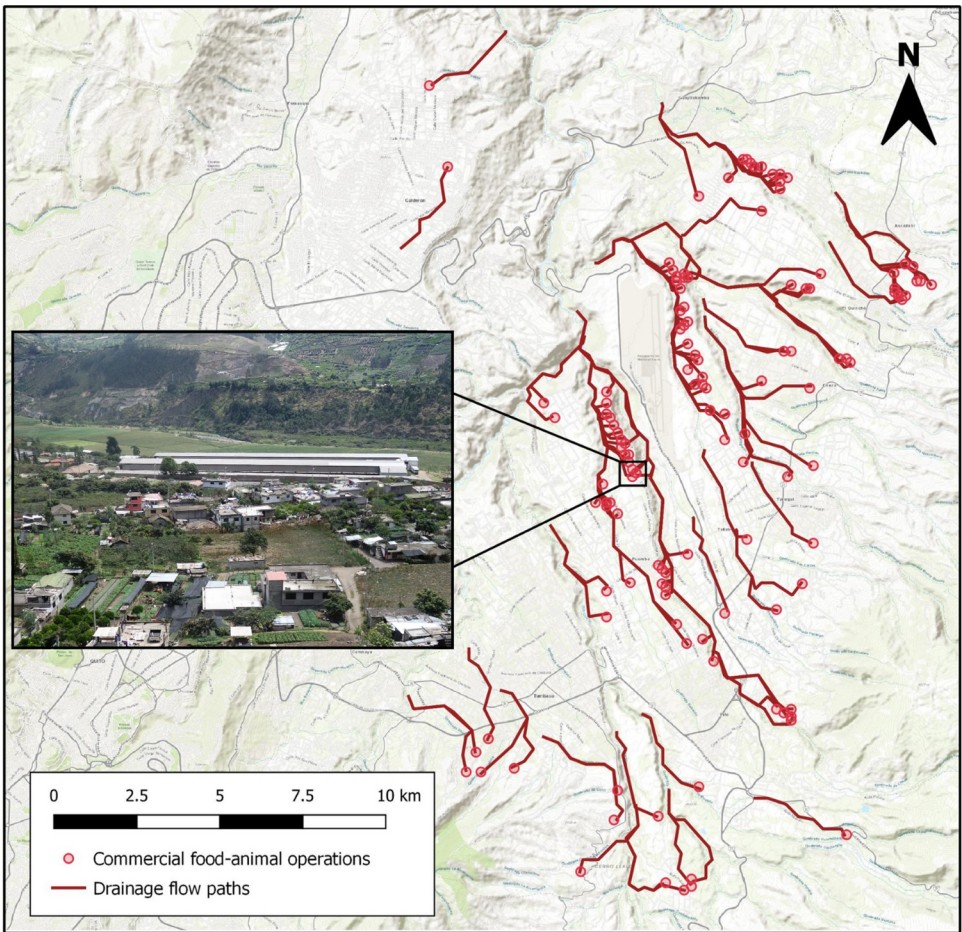

**Fig 1. Map of active commercial food animal production operations (*n* = 130) and their drainage flow paths in the study area, east of Quito, Ecuador.** The inset is an aerial photograph of commercial poultry production facilities in the study area (image credit: Jay P. Graham). Map created in QGIS; contains information from OpenStreetMap and OpenStreetMap Foundation, which is made available under the Open Database License.

Household-level exposure to small-scale food animal production was defined as household ownership of any food animals (i.e., chickens, pigs, cattle, sheep, goats, ducks, guinea pigs, rabbits, or quail). Household surveys captured information on caregiver-reported household ownership of food animals and other domestic animals, as well as household characteristics, caregiver and child demographics, health status, and other potential risk factors. Surveys were created using Open Data Kit (ODK) Build and trained enumerators used ODK Collect on Android devices for mobile data collection [42]. Encrypted survey forms were sent to ODK Aggregate on a secure server at USFQ upon completion and were subsequently downloaded for de-identification and analysis.

## Outcome assessment

A single stool sample from the youngest eligible child per household was collected at each visit to determine enteric carriage of 3GCR-EC and ESBL production based on phenotypic testing, described below. Caregivers were given supplies to collect a child stool sample, which were double-bagged and stored in a fridge or on ice (4˚C), and the enumerator returned to collect the sample the next day. In the case that a child had not defecated before the enumerator returned, the enumerator came back to the household the following day to collect the sample. If domestic or food animals were present at the household at the time of data collection, a single stool sample per animal species was collected from the environment where the animals defecate. Fecal samples were placed in sterile containers and stored on ice (4˚C) during transportation to the microbiology lab at USFQ. Samples were processed at USFQ within 5 h of collection to identify 3GCR-EC and conduct antibiotic susceptibility testing.

## Microbiological methods

To screen for 3GCR-EC and improve sensitivity for detecting ESBL production among both dominant and nondominant strains of *E. coli*, fecal samples were plated on MacConkey agar (Difco, Sparks, Maryland) with 2 mg/L of ceftriaxone and incubated for 18 h at 37˚C [43]. Up to 5 ceftriaxone-resistant isolates phenotypically matching *E. coli* were selected from each fecal sample and preserved at −80˚C in Trypticase Soy Broth medium (Difco, Sparks, Maryland) with 20% glycerol. 3GCR-EC isolates for each fecal sample were thawed and regrown on MacConkey agar at 37˚C for 18 to 24 h for evaluation of antibiotic susceptibility by the disk diffusion method (Kirby Bauer test) on Mueller–Hinton agar (Difco, Sparks, Maryland). To confirm presumptive *E. coli* isolates, colonies were inoculated onto Chromocult coliform agar (Merck, Darmstadt, Germany).

## Antibiotic susceptibility testing

Antibiotic susceptibility testing of all 3GCR-EC isolates was conducted for 10 antibiotics: ampicillin (AM; 10 μg), ceftazidime (CAZ; 30 μg), ciprofloxacin (CIP; 5 μg), cefotaxime (CTX; 30 μg), cefazolin (CZ; 30 μg), cefepime (FEP; 30 μg), gentamicin (GM; 10 μg), imipenem (IPM; 10 μg), trimethoprim/sulfamethoxazole (SXT; 1.25 per 23.75 μg), and tetracycline (TE; 30 μg). Isolates were identified as either susceptible or resistant to each antibiotic according to the resistance or susceptibility interpretation criteria from Clinical and Laboratory Standards Institute (CLSI) guidelines [44]. *E. coli* ATCC 25922 was used as the quality control strain. Multidrug resistance (resistant to 3 or more classes) was determined based on the number of macro-classes to which each isolate was resistant. Macro-classes were defined as cephalosporin/beta-lactamase inhibitors, penicillins, aminoglycosides, carbapenems, fluoroquinolones, tetracyclines, and folate pathway inhibitors.

For phenotypic confirmation of ESBL production, the combination disk diffusion test was used with CAZ and CAZ/CLA (ceftazidime with clavulanic acid) as outlined in the CLSI guidelines [44]. In the first 4 cycles of data collection, up to 5 *E. coli* isolates per sample were selected and preserved for analysis. 3GCR-EC isolates from the same fecal sample with identical phenotypic resistance profiles were considered duplicates and were de-duplicated prior to analyses. Due to limited laboratory resources after the SARS-CoV-2 pandemic began in 2020 and the high rate of clonal relationships between *E. coli* isolated from the same sample (based on preliminary sequencing), only 1 *E. coli* isolate per sample was selected and preserved for analysis during the fifth cycle of data collection.

## DNA sequencing and analysis

Genomic DNA was extracted from the isolates using Wizard Genomic DNA Purification (Promega) kits and QIAGEN DNEasy Blood & Tissue Kits according to the manufacturer's instructions. We conducted quality control of extracted DNA prior to library creation at the University of Minnesota Genomics Center, including PicoGreen quantification, quantitative capillary electrophoretic sizing (Agilent), and functional quantification (KAPA Biosystems qPCR). Whole-genome sequencing was carried out at the University of Minnesota. In brief, we sequenced whole-genome *E. coli* isolates using either Illumina MiSeq or NovaSeq with Nextera XT libraries. Following sequencing, raw reads were quality-trimmed and adapter-trimmed using trimmomatic [45]. Assemblies of reads was performed using SPAdes [46], then ARGs were identified using ABRicate (version 0.8.13) and a curated version of the ResFinder database [47]. We also performed in silico multilocus sequence typing (MLST) based on 7 housekeeping genes (adk, fumC, gyrB, icd, mdh, purA, and recA), an additional 8 housekeeping genes (dinB, icdA, pabB, polB, putP, trpA, trpB, and uidA), and core genome (cgMLST) using MLST 2.0 [48] and cgMLSTFinder 1.1 [49]. Detailed methods are previously described [50]. Raw reads from isolates sequenced in this study are available at the NCBI Short Read Archive (SRA) under BioProject accession no. PRJNA861272.

## Statistical analyses

Multivariable log-binomial regression models were used to estimate unadjusted and adjusted relative risks (RRs) for the associations between commercial or household food animal exposures and 3GCR-EC and ESBL-EC carriage (i.e., main effects). To control for confounding and strong predictors of the outcomes variables, all adjusted models included prespecified covariates (child age, child sex, child antibiotic use in the past 3 months, household asset score as a proxy for socioeconomic status, and caregiver education level) identified using a directed acyclic graph and existing literature (Fig A in S1 Files) [51]. Next, we included an interaction term in adjusted models, comparing combined effects of household and commercial food animal exposures to a single referent group (unexposed to both commercial and household food animals). We also assessed effect measure modification by estimating the effect of each measure of exposure to commercial food animals among those with versus without food animals (i.e., stratum-specific effects). Estimates and *P*-values from interaction models were used to determine multiplicative interaction based on an alpha level of <0.10. Finally, we ran additional log-binomial regression models to estimated associations between secondary risk factors of interest related to household animal ownership, animal contact and feces management, and presence of 3GCR-/ESBL-EC in animal stool collected at households. Exposures were treated as binary and categorical using cut-points selected based on the data distribution and interpretability. The outcomes, 3GCR-EC and ESBL-EC carriage, were treated as binary and modelled at the isolate level, adjusting for clustering at the individual/household level and

estimating robust standard errors with generalized estimating equations. In a sensitivity analysis, we reproduced this main analysis using only the first *E. coli* isolate per fecal sample to assess the potential for bias due to the change in number of isolates and probability of detecting the outcomes at each time point. Individual observations with missing exposure, outcome, or covariate data were removed for this complete case analysis. Statistical analyses and visualizations were completed in R version 3.6.1 [38] using the *dplyr* [52], *tableone* [53], *ggplot2* [54], *geepack* [55], *multcomp* [56], and *car* [57] packages. Exposure and outcome data used in epidemiological analyses are published and publicly available on Dryad (https://doi.org/10.5061/dryad.41ns1rnm7).

## Results

A total of 605 households across 7 semirural parishes east of Quito, Ecuador were enrolled throughout the study period between July 2018 and September 2021. We enrolled 374 households in the initial cycle of data collection and recruited new households (using the same enrollment criteria) to enroll in subsequent data collection cycles to account for loss to follow-up. During the fourth cycle, data collection was halted in March 2020 due to lockdown restrictions during the SARS-CoV-2 pandemic, resulting in significant loss to follow-up (39.7%, 151/380 lost to follow-up). However, in the fifth cycle in 2021, we re-enrolled 63.4% (241/380) of participants from cycle 3 and 78.8% (186/236) of participants from cycle 4 (Fig B in S1 Files). A total of 1,739 child fecal samples were collected throughout the study period, from which 920 distinct 3GCR-EC colonies were isolated. Two percent (40/1,739) of child fecal samples were missing corresponding survey data, resulting in a total of 910 3GCR-EC isolates from 1,699 child fecal samples across 600 children from different households. The median number of household visits (i.e., fecal samples) for each child was 3; 23% of the 600 households were visited 5 times, 18% 4 times, 11% 3 times, 11% 2 times, and 37% were visited once. Overall, 1.8% (11/605) of all enrolled households were missing either exposure, outcome, or covariate data and were not included in statistical analyses. After removing children with missing data, 904 3GCR-EC isolates from 1,677 child fecal samples from 594 children remained. This resulted in a total of 1,940 observations (including multiple isolates per fecal sample) in the final dataset for the primary statistical analysis.

### Household and child characteristics

In a majority of the 594 included households, most primary caregivers had at least a high school or college level education, ranging from 67.1% to 79.5% across the data collection cycles (Table 1). The average age of children participating in the study was 1.8 years during the first cycle of data collection (Table 2). Access to drinking water and sanitation was high; 98.8% of households had a flush toilet (to sewer or septic tank), 92.2% had piped drinking water inside their home, and 98.6% had 24-h access to drinking water (Table M in S1 Files). Over 93% of households had household handwashing stations with the presence of both soap and water, confirmed by observation (Table 1). However, caregiver-reported child handwashing frequency suggested that most children rarely (44.4% to 69.6%) washed their hands after contact with animals (Table 2). Over the course of the entire study period, 36.7% of caregivers reported that their child played near animal feces in the last 3 weeks, 36.0% reported that their child had contact with livestock at least once per week in the last 3 months, and 67.9% reported child contact with pets at least once per week in the last 3 months. Seven-day caregiver-reported child diarrhea prevalence was 20.3% at the beginning of the study period in 2018 and declined to 4.2% at the final data collection cycle in 2021 during the SARS-CoV-2 pandemic (Table 2). Similarly, treatment for infection in the last 3 months declined from 31.1% at the first cycle to

**Table 1. Characteristics of study households in semirural Quito, Ecuador at each cycle of data collection between July 2018 and September 2021.**

| | Data collection cycle | | | | |
|---|---|---|---|---|---|
| | 1<br>n (%) | 2<br>n (%) | 3<br>n (%) | 4<br>n (%) | 5<br>n (%) |
| *Total* | 370 (100) | 358 (100) | 365 (100) | 225 (100) | 356 (100) |
| *Parish* | | | | | |
| El Quinche | 20 (5.4) | 17 (4.7) | 30 (8.2) | 16 (7.1) | 26 (7.3) |
| Puembo | 20 (5.4) | 21 (5.9) | 37 (10.1) | 16 (7.1) | 23 (6.5) |
| Pifo | 86 (23.2) | 77 (21.5) | 75 (20.5) | 55 (24.4) | 75 (21.1) |
| Tababela | 12 (3.2) | 9 (2.5) | 8 (2.2) | 6 (2.7) | 13 (3.7) |
| Tumbaco | 23 (6.2) | 35 (9.8) | 24 (6.6) | 17 (7.6) | 22 (6.2) |
| Yaruqui | 146 (39.5) | 146 (40.8) | 141 (38.6) | 80 (35.6) | 141 (39.6) |
| Missing | 4 (1.1) | 0 (0.0) | 0 (0.0) | 0 (0.0) | 0 (0.0) |
| *Caregiver education level* | | | | | |
| High school or college | 276 (74.6) | 263 (73.5) | 256 (70.1) | 151 (67.1) | 283 (79.5) |
| Elementary | 93 (25.1) | 95 (26.5) | 108 (29.6) | 68 (30.2) | 69 (19.4) |
| Missing | 1 (0.3) | 0 (0.0) | 1 (0.3) | 6 (2.7) | 4 (1.1) |
| *Household size (mean (SD))* | 4.5 (1.4) | 4.6 (1.5) | 4.6 (1.4) | 4.4 (1.2) | 4.4 (1.5) |
| *Household water treatment* | | | | | |
| No treatment | 194 (52.4) | 170 (47.5) | 223 (61.1) | 138 (61.3) | 229 (64.3) |
| Boil | 159 (43.0) | 154 (43.0) | 119 (32.6) | 68 (30.2) | 109 (30.6) |
| Chlorinate or filter | 3 (0.8) | 5 (1.4) | 5 (1.3) | 3 (1.3) | 2 (0.6) |
| Other | 14 (3.8) | 29 (8.1) | 18 (4.9) | 16 (7.1) | 16 (4.5) |
| *Household handwashing station* | | | | | |
| Soap and water | 354 (95.7) | 344 (96.1) | 347 (95.1) | 211 (93.8) | 340 (95.5) |
| Soap or water, only | 9 (2.4) | 14 (4.0) | 17 (4.7) | 14 (6.2) | 16 (4.5) |
| Neither | 6 (1.6) | 0 (0.0) | 1 (0.3) | 0 (0.0) | 0 (0.0) |
| Missing | 1 (0.3) | 0 (0.0) | 0 (0.0) | 0 (0.0) | 0 (0.0) |
| *Caregiver worked with animals in past 6 months*[a] | 105 (28.4) | 79 (22.1) | 80 (21.9) | 73 (32.4) | 113 (31.7) |
| *Household owns animals* | 233 (63.0) | 241 (67.3) | 244 (66.8) | 157 (69.8) | 236 (66.3) |
| Missing | 3 (0.8) | 1 (0.3) | 0 (0.0) | 0 (0.0) | 0 (0.0) |
| *Household owns food animals* | 110 (29.7) | 117 (32.7) | 107 (29.3) | 63 (28.0) | 92 (25.8) |
| Missing | 2 (0.5) | 1 (0.3) | 0 (0.0) | 0 (0.0) | 0 (0.0) |
| *Household food animals (mean (SD))* | 6.9 (19.3) | 5.7 (14.1) | 6.9 (19.1) | 9.2 (39.4) | 5.9 (21.3) |
| *Antibiotic use for household food animals* | 29 (26.4) | 22 (18.8) | 11 (10.3) | 6 (9.5) | 0 (0.0) |
| Missing | 7 (6.4) | 4 (3.4) | 1 (0.9) | 0 (0.0) | 9 (9.8) |
| *Distance to nearest commercial food animal operation* | | | | | |
| ≥1.5 km | 155 (41.9) | 162 (45.3) | 154 (42.2) | 97 (43.1) | 170 (47.8) |
| 1–1.49 km | 90 (24.3) | 86 (24.0) | 87 (23.8) | 52 (23.1) | 70 (19.7) |
| 0.5–0.9 km | 60 (16.2) | 54 (15.1) | 73 (20.0) | 45 (20.0) | 71 (19.9) |
| <0.5 km | 61 (16.5) | 56 (15.6) | 51 (14.0) | 31 (13.8) | 45 (12.6) |
| Missing | 4 (1.1) | 0 (0.0) | 0 (0.0) | 0 (0.0) | 0 (0.0) |
| *Commercial food animal operations within 5 km radius* | | | | | |
| ≤5 | 115 (31.1) | 118 (33.0) | 113 (31.0) | 71 (31.6) | 203 (57.0) |
| 6–10 | 93 (25.1) | 89 (24.9) | 84 (23.0) | 52 (23.1) | 44 (12.4) |
| 11–20 | 88 (23.8) | 90 (25.1) | 76 (20.8) | 51 (22.7) | 47 (13.2) |
| >20 | 70 (18.9) | 61 (17.0) | 92 (25.2) | 51 (22.7) | 62 (17.4) |
| Missing | 4 (1.1) | 0 (0.0) | 0 (0.0) | 0 (0.0) | 0 (0.0) |
| *Distance to drainage path of commercial food animal operation* | | | | | |
| >500 m | 154 (41.6) | 161 (45.0) | 150 (41.1) | 99 (44.0) | 167 (46.9) |

*(Continued)*

**Table 1.** (Continued)

| | Data collection cycle | | | | |
|---|---|---|---|---|---|
| 101–500 m | 158 (42.7) | 133 (37.2) | 162 (44.4) | 92 (40.9) | 138 (38.8) |
| ≤100 m | 60 (16.2) | 67 (18.7) | 59 (16.2) | 39 (17.3) | 56 (15.7) |
| Missing | 4 (1.1) | 0 (0.0) | 0 (0.0) | 0 (0.0) | 0 (0.0) |

a Including working with live animals, with animal feces, or in meat processing.

8.1% at the fifth cycle; child antibiotic use in the last 3 months also declined from 26.2% at cycle 1 to 5.9% at cycle 5 (Table 2).

## Food animal production and domestic animals

We identified 130 active commercial food animal operations in our study site including 122 poultry facilities, 5 hog facilities, 2 horse facilities, and 1 milk production facility. Across the 7

**Table 2. Characteristics and behaviors of children in study households at each cycle of data collection between July 2018 and September 2021.**

| | Data collection cycle | | | | |
|---|---|---|---|---|---|
| | 1 n (%) | 2 n (%) | 3 n (%) | 4 n (%) | 5 n (%) |
| Total | 370 (100) | 358 (100) | 365 (100) | 225 (100) | 356 (100) |
| Child age in years (mean (SD)) | 1.8 (1.3) | 2.0 (1.3) | 2.4 (1.4) | 3.0 (1.3) | 3.8 (1.6) |
| Child sex | | | | | |
| Female | 171 (46.2) | 162 (45.3) | 154 (42.2) | 96 (42.7) | 158 (44.4) |
| Male | 199 (53.8) | 196 (54.7) | 211 (57.8) | 129 (57.3) | 198 (55.6) |
| Child contact with livestock in last 3 months | | | | | |
| Never | 248 (67.0) | 254 (70.9) | 238 (65.2) | 126 (56.0) | 205 (57.6) |
| <1 time per week | 42 (11.4) | 26 (7.3) | 54 (14.8) | 14 (6.2) | 25 (7.0) |
| 1–2 times per week | 31 (8.4) | 32 (8.9) | 39 (10.7) | 50 (22.2) | 50 (14.0) |
| 3+ times per week | 49 (13.2) | 46 (12.8) | 34 (9.3) | 35 (15.6) | 75 (21.1) |
| Missing | 0 (0.0) | 0 (0.0) | 0 (0.0) | 0 (0.0) | 1 (0.3) |
| Child contact with pets in last 3 months | | | | | |
| Never | 132 (35.7) | 122 (34.1) | 108 (29.6) | 64 (28.4) | 112 (31.5) |
| <1 time per week | 54 (14.6) | 34 (9.5) | 65 (17.8) | 12 (5.3) | 23 (6.5) |
| 1–2 times per week | 52 (14.1) | 50 (14.0) | 73 (20.0) | 59 (26.2) | 50 (14.0) |
| 3+ times per week | 132 (35.7) | 152 (42.5) | 119 (32.6) | 90 (40.0) | 171 (48.0) |
| Child played near animal feces in last 3 weeks | 115 (31.1) | 99 (27.7) | 148 (40.5) | 77 (34.2) | 175 (49.2) |
| Missing | 2 (0.5) | 0 (0.0) | 0 (0.0) | 0 (0.0) | 3 (0.8) |
| Child handwashing after contact with animals | | | | | |
| Never | 43 (11.6) | 13 (3.6) | 4 (1.1) | 3 (1.3) | 3 (0.8) |
| Rarely | 185 (50.0) | 249 (69.6) | 245 (67.1) | 129 (57.3) | 158 (44.4) |
| Sometimes | 83 (22.4) | 68 (19.0) | 109 (29.9) | 87 (38.7) | 159 (44.7) |
| Always | 18 (4.9) | 9 (2.5) | 7 (1.9) | 6 (2.7) | 36 (10.1) |
| Do not know/does not apply | 41 (11.1) | 19 (5.3) | 0 (0.0) | 0 (0.0) | 0 (0.0) |
| Child had diarrhea in last 7 days | 75 (20.3) | 77 (21.5) | 43 (11.8) | 36 (16.0) | 15 (4.2) |
| Missing | 2 (0.5) | 1 (0.3) | 1 (0.3) | 0 (0.0) | 2 (0.6) |
| Child treated for infection in last 3 months | 115 (31.1) | 98 (27.4) | 84 (23.0) | 54 (24.0) | 29 (8.1) |
| Missing | 3 (0.8) | 1 (0.3) | 0 (0.0) | 0 (0.0) | 0 (0.0) |
| Child took antibiotics in last 3 months | 97 (26.2) | 67 (18.7) | 63 (17.3) | 26 (11.6) | 21 (5.9) |
| Missing | 0 (0.0) | 2 (0.6) | 0 (0.0) | 0 (0.0) | 0 (0.0) |

parishes in the study site, 4 parishes (Tababela, Tumbaco, Checa (Chilpa), and Pifo) had low-intensity commercial production with fewer than 10 commercial food animal operations, each. Three parishes (El Quinche, Puembo, and Yaruqui) had high-intensity production, with more than 30 commercial food animal operations, each (Table F in S1 Files). More than half of study households were located <1.5 km from a commercial food animal operation (55.7%) or were within 500 m of a commercial operation's drainage flow path (56.1%) throughout the study period. During the first 4 cycles of data collection, at least 67% of households were within 5 km of 6 or more commercial food animal operations; this dropped to 43% in the fifth cycle (Table 1).

We collected 1,871 animal fecal samples from 376 matched households and identified 1,060 3GCR-EC isolates to assess as a potential risk factor. Throughout the study period, 66.4% of caregivers reported owning any type of animal, with 29.2% owning food animals and an average of 5.7 to 9.2 food animals at each data collection cycle (Table 1). Among households that owned food animals, reported antibiotic use in food animals was low, declining from 26.4% at the beginning of the study period to 0% at the end of the study (Table 1). Thirty-two percent of households owned backyard chickens, 17% owned guinea pigs, 11% owned pigs, 7% owned cattle, and 3% owned goats or sheep. The average flock size for backyard chickens was 13.7 (standard deviation (SD): 29.9). There was an average of 16 guinea pigs (SD: 19.6), 4.3 pigs (SD: 6.6), 2.8 cows (SD: 2.6), 2.9 goats (SD: 3.2), and 4.0 sheep (SD: 6.3). Eighty percent of households owned dogs, with an average of 2.3 dogs per household (SD: 1.6).

## Characterization of 3GCR-EC isolates

Sixty-one percent of children ($n = 365/600$) were carriers of 3GCR-EC and 25% ($n = 149/600$) were carriers of ESBL-EC at least once throughout the study period. 3GCR-EC were detected in 38% of child samples ($n = 652/1,699$) and 51% of animal samples ($n = 959/1,871$) from matched households. Of 910 3GCR-EC isolated from child samples, 36% were resistant to fourth-generation cephalosporin, cefepime, in phenotypic susceptibility testing (Table B in S1 Files). Eighty-six percent of child 3GCR-EC isolates were multidrug-resistant (3 or more classes), 37% were extensively drug-resistant (5 or more classes), and 22% were phenotypic ESBL producers (Table B in S1 Files). Notably, 5 children (<1%) from different households had 1 *E. coli* isolate that was phenotypically resistant to the carbapenem drug, imipenem (Table B in S1 Files). All of these imipenem-resistant isolates were multidrug-resistant to 4 or more drug classes, with each isolate also expressing resistance to ampicillin and cephalosporin antibiotics. Phenotypic resistance to antibiotics among 3GCR-EC isolates from animal fecal samples are described in Table C in S1 Files.

We analyzed whole-genome sequencing data for a subset of 571 3GCR-EC isolated from fecal samples of the 365 children that were 3GCR-EC carriers. We first selected 1 isolate per child for sequencing, then aimed to randomly select an additional 200 isolates to include more ESBL-EC in sequencing analyses. The most prevalent sequence type (ST) among these isolates was ST 10 (7%, 40/571), while other clinically important STs like ST 131 and ST 117 accounted for about 3% of sequenced isolates (Table D in S1 Files). 3GCR-EC isolates had a mean of 9.7 total ARGs (SD: 4.3). Overall, the proportion of 3GCR-EC isolates with *bla* genes was 65%, 52%, 8%, 5%, and 5% for $bla_{CTX-M}$-encoding, $bla_{TEM}$, $bla_{CMY}$, $bla_{OXA}$, and $bla_{SHV}$ genes, respectively. Ten (2%) 3GCR-EC isolates had an *mcr-1* gene, indicating resistance to the last-line antibiotic colistin. Eighty-two percent ($n = 119/145$) of phenotypic ESBL-EC and 77% ($n = 330/426$) of non-ESBL-producing 3GCR-EC carried at least 1 *bla* gene. Among phenotypic ESBL-EC, the most prevalent *bla* genes were $bla_{CTX-M-55}$, $bla_{TEM-141}$, $bla_{TEM-1B}$, $bla_{CTX-M-15}$, and $bla_{OXA-1}$ (Table E in S1 Files). Among the most prevalent CTX-M-encoding

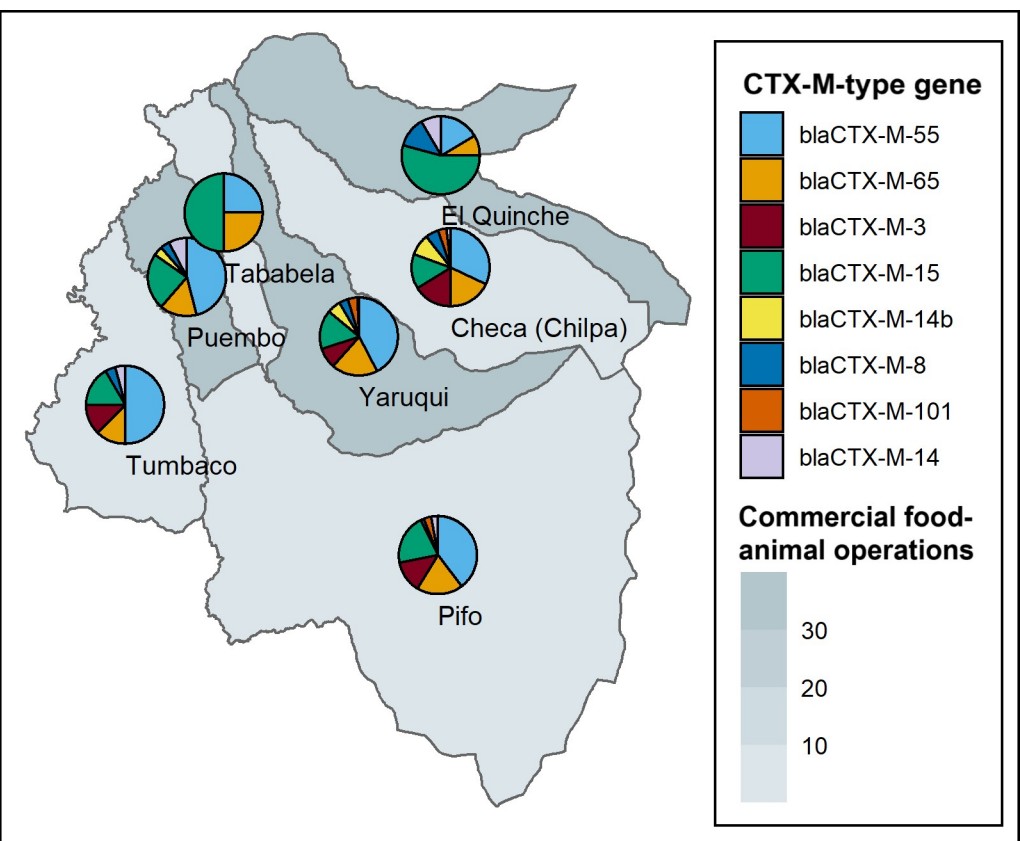

**Fig 2. Map of distribution of CTX-M-type genes from a subset of 571 3GCR-EC isolated from children in 7 parishes east of Quito, Ecuador.** Pie chart sections represent the proportion of isolates with a specific CTX-M-type gene detected, among those represented in the plot legend. Shaded areas represent parishes, where darker shading indicates a higher number of commercial food animal operations in a given parish. Map created in RStudio; Ecuador parish boundary shapefiles are made publicly available by the INEC and the United Nations OCHA. Data and license information are available at https://data.humdata.org/dataset/cod-ab-ecu?. 3GCR-EC, third-generation cephalosporin-resistant *E. coli*; INEC, Instituto Nacional de Estadística y Censos; OCHA, Office for the Coordination of Humanitarian Affairs.

genes, the most prevalent genes in parishes with high-intensity commercial food animal production were $bla_{CTX-M-55}$, $bla_{CTX-M-65}$, $bla_{CTX-M-15}$, and $bla_{CTX-M-14}$ (Fig 2 and Table H in S1 Files). Parishes with lower intensity commercial food animal production had a higher prevalence of $bla_{CTX-M-3}$, though $bla_{CTX-M-55}$, $bla_{CTX-M-65}$, and $bla_{CTX-M-15}$ were also detected in these parishes (Fig 2 and Table H in S1 Files).

## Risk factors for 3GCR-EC

The prevalence of 3GCR-EC was higher in all food animal exposure groups compared to the unexposed, where the denominators include all fecal samples and isolates within each exposure group (Fig 3). In models assessing main effects of household and commercial food animal exposures, the density of commercial operations (>5 in 5 km radius) was the only exposure associated with an increased risk of 3GCR-EC (RR: 1.27; 95% CI: 1.11, 1.46) (Table 3). In models adjusting for interaction between household and commercial food animal exposures, children with >5 commercial food animal operations in a 5-km radius of their household had 1.36 times the risk (95% confidence interval (CI): 1.16, 1.59) of 3GCR-EC carriage than those with ≤5 operations within 5 km when controlling for household food animal ownership

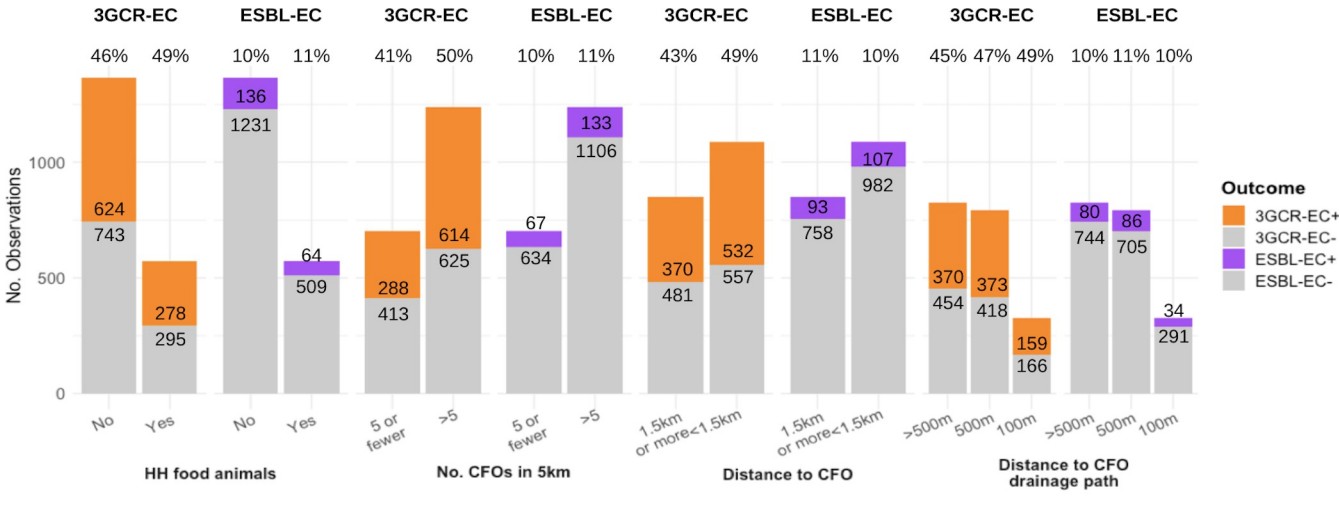

**Fig 3. Bar plot of distribution of 3GCR-EC and ESBL-EC carriage stratified by exposures to HH food animals and CFOs.** Number of observations includes fecal samples that were negative for 3GCR-EC (light gray bars for 3GCR-EC), isolates from fecal samples that were 3GCR-EC (orange bars), or ESBL-EC (purple bars) based on phenotypic susceptibility testing, and isolates plus fecal samples negative for both 3GCR-EC and ESBL-EC (light gray bars for ESBL-EC). Percent of observations positive for each outcome are listed above each bar. 3GCR-EC, third-generation cephalosporin-resistant *E. coli*; CFO, commercial food animal operation; ESBL-EC, extended-spectrum beta-lactamase *E. coli*; HH, household.

(Table 4). However, among those with household food animals, there was no effect of commercial operation density on 3GCR-EC carriage (RR: 1.05; 95% CI: 0.83, 1.33) (Table 4). There was not an excess risk of being exposed to both household food animals and >5 commercial operations (relative excess risk due to interaction (RERI): −0.29, 95% CI: −0.64, 0.06) (Table 4). However, the combination of owning household food animals and living <1.5 km from the nearest commercial operation was associated—with borderline significance—with an

**Table 3. Unadjusted and adjusted RRs of 3GCR-EC and ESBL-EC for main effects of household food animal ownership and exposures to commercial food animal production.**

| | 3GCR-EC | | | | ESBL-EC | | | |
|---|---|---|---|---|---|---|---|---|
| | Unadjusted RR (95% CI) | P-Value | Adjusted RR (95% CI) | P-Value | Unadjusted RR (95% CI) | P-Value | Adjusted RR (95% CI) | P-Value |
| Household owns food animals (ref = no)[a] | 1.06 (0.94, 1.19) | *0.388* | 1.04 (0.92, 1.17) | *0.521* | 1.12 (0.82, 1.52) | *0.472* | 1.10 (0.81, 1.51) | *0.541* |
| > 5 CFO in 5 km radius (ref = ≤5)[b] | 1.26 (1.10, 1.45) | *0.001* | 1.27 (1.11, 1.46) | *0.001* | 1.12 (0.84, 1.51) | *0.444* | 1.14 (0.84, 1.53) | *0.404* |
| < 1.5 km to nearest CFO (ref = ≥ 1.5 km)[b] | 1.21 (0.84, 1.75) | *0.305* | 1.10 (0.98, 1.25) | *0.118* | 1.24 (0.60, 2.60) | *0.561* | 0.91 (0.68, 1.22) | *0.520* |
| 101–500 m to CFO drainage path (ref = >500 m)[b] | 1.05 (0.91, 1.21) | *0.486* | 1.04 (0.91, 1.20) | *0.563* | 1.14 (0.83, 1.57) | *0.426* | 1.15 (0.83, 1.59) | *0.392* |
| 0–100 m to CFO drainage path (ref = >500 m)[b] | 1.13 (0.96, 1.33) | *0.132* | 1.08 (0.92, 1.27) | *0.331* | 1.15 (0.77, 1.71) | *0.510* | 1.15 (0.77, 1.72) | *0.499* |

All RRs estimated with log-binomial regression models using generalized estimating equations to adjust for repeated measures and estimate robust 95% CI. Adjusted RR included the following covariates: caregiver education, asset score, child age and sex, and child antibiotic use in the last 3 months. *N* = 1,940 observations across 1,677 child fecal samples (including 910 total 3GCR-EC isolates) for 594 children. CFO = commercial food animal operation. RR: relative risk. CI: confidence interval. 3GCR-EC: third-generation cephalosporin-resistant *E. coli*. ESBL-EC: extended-spectrum beta-lactamase producing *E. coli*.

[a] Adjusted models also controlled for number of commercial food animal productions in 5km radius.

[b] Adjusted models also controlled for household food animal ownership.

**Table 4. Adjusted RRs of 3GCR-EC carriage among children given combined exposures to commercial food animal production and effect measure modification by household food animal ownership.**

| | | Adjusted RR for 3GCR-EC (95% CI) | | Interaction *P*-value | RERI (95% CI) |
|---|---|---|---|---|---|
| | | No household food animals | Household food animals | | |
| **No. CFOs in 5 km radius** | ≤5 | 1.00 (ref) | 1.22 (0.95, 1.55) | - | - |
| | >5 | 1.36 (1.16, 1.59) | 1.28 (1.06, 1.54) | 0.077 | −0.29 (−0.64, 0.06) |
| | >5 within strata of household food animals | 1.36 (1.16, 1.59) | 1.05 (0.83, 1.33) | - | - |
| **Distance to nearest CFO** | ≥1.5 km | 1.00 (ref) | 1.02 (0.83, 1.26) | - | - |
| | <1.5 km | 1.09 (0.95, 1.27) | 1.15 (0.98, 1.36) | 0.827 | 0.03 (−0.23, 0.30) |
| | <1.5 km within strata of household food animals | 1.09 (0.95, 1.27) | 1.13 (0.91, 1.39) | - | - |
| **Distance to nearest CFO drainage path** | >500 m | 1.00 (ref) | 0.98 (0.80, 1.21) | - | - |
| | 101–500 m | 1.00 (0.85, 1.18) | 1.16 (0.90, 1.49) | 0.327 | 0.16 (−0.14, 0.45) |
| | ≤100 m | 1.10 (0.91, 1.34) | 1.07 (0.82, 1.38) | 0.818 | −0.04 (−0.36, 0.28) |
| | 101–500 m within strata of household food animals | 1.00 (0.85, 1.18) | 1.14 (0.93, 1.39) | - | - |
| | ≤100 m within strata of household food animals | 1.10 (0.91, 1.34) | 1.05 (0.84, 1.31) | - | - |

Log-binomial regression models with generalized estimating equations included interaction terms between commercial and household food animal exposure variables, and included the following covariates: caregiver education, asset score, child age and sex, and child antibiotic use in the last 3 months. *N* = 1,940 observations across 1,677 child fecal samples (including 910 total 3GCR-EC isolates) for 594 children. CFO = commercial food animal operation. RR: relative risk. CI: confidence interval. 3GCR-EC: third-generation cephalosporin-resistant *E. coli*. RERI: relative excess risk due to interaction.

increased risk in 3GCR-EC carriage (RR: 1.15; 95% CI: 0.98, 1.36) compared to those without food animals who lived further from commercial food animal operations (Table 4). Proximity to a drainage flow path from a commercial food animal operation was not associated with 3GCR-EC carriage, regardless of household food animal ownership (Table 4). These results were largely robust to sensitivity analyses (Table I in S1 Files). Other risk factors for 3GCR-EC among children included child pet contact in the last 3 months (RR: 1.23; 95% CI: 1.09, 1.39) and pig ownership (RR: 1.23; 95% CI: 1.02, 1.48) (Fig 4) (Table J in S1 Files).

## Risk factors for ESBL-EC

We did not detect significant associations or interactions between the density of or proximity to commercial food animal operations, household food animal ownership, and ESBL-EC carriage. In models adjusting for interactions between exposures, household food animal ownership and increased proximity to a drainage path were not significantly associated with an increased risk of ESBL-EC carriage among children. However, the effect sizes were larger for combined and increasing intensity of exposures. When controlling for household food animal ownership, the RR for ESBL-EC was 1.10 (95% CI: 0.76, 1.60) for those living 101 to 500 m from a drainage path and was 1.32 (95% CI: 0.71, 2.46) for those 101 to 500 m from a drainage path compared to those >500 m (Table 5). The RR further increased to 1.80 (95% CI: 0.94, 3.45) for those living within 100 m of a drainage path and with household food animals compared to those >500 m from a drainage path and without household food animals; this

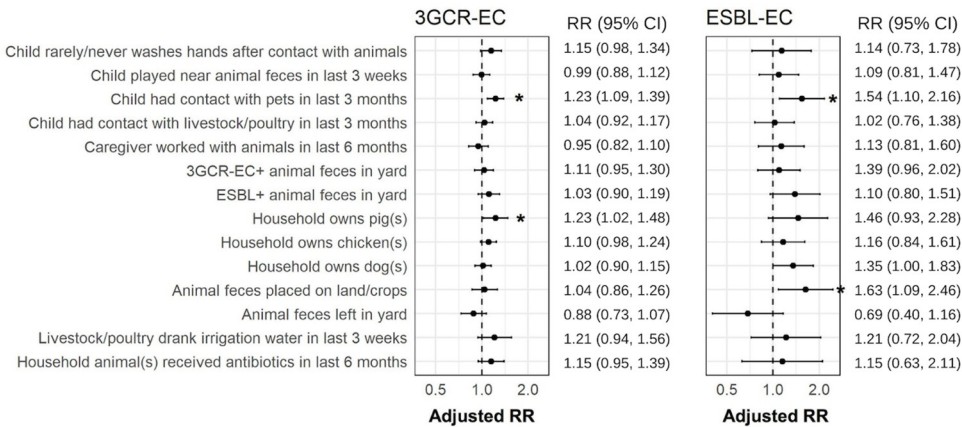

**Fig 4. Additional risk factors for 3GCR-EC and ESBL-EC carriage among children in semirural parishes of Quito, Ecuador.** Points are adjusted RR and error bars are 95% CIs; asterisks (*) indicate significance given *alpha* = 0.05 (corresponding data including sample sizes and *P*-values in Tables K and J in S1 Files). RR are adjusted for repeated measures and controlled for the following covariates: caregiver education, asset score, child age and sex, and child antibiotic use in the last 3 months. 3GCR-EC, third-generation cephalosporin-resistant *E. coli*; CI, confidence interval; ESBL-EC, extended-spectrum beta-lactamase *E. coli*; RR, relative risk.

**Table 5. Adjusted RRs of ESBL-EC carriage among children given combined exposures to commercial food animal production and household food animals, including both interaction effects (effects of individual and combined exposures vs. no exposures) and stratum-specific effects (effects of commercial food animal exposures within strata of household food animal ownership vs. no exposures).**

| | | Adjusted RR for ESBL-EC (95% CI) | | Interaction *P*-value | RERI (95% CI) |
|---|---|---|---|---|---|
| | | No household food animals | Household food animals | | |
| **No. CFOs in 5 km radius** | ≤5 | 1.00 (ref) | 1.18 (0.69, 2.01) | - | - |
| | >5 | 1.17 (0.82, 1.68) | 1.24 (0.82, 1.86) | 0.737 | −0.12 (−0.89, 0.66) |
| | >5 within strata of household food animals | 1.17 (0.82, 1.68) | 1.05 (0.61, 1.81) | - | - |
| **Distance to nearest CFO** | ≥1.5 km | 1.00 (ref) | 0.91 (0.57, 1.46) | - | - |
| | <1.5 km | 0.81 (0.57, 1.16) | 1.05 (0.69, 1.59) | 0.268 | 0.32 (−0.23, 0.88) |
| | <1.5 km within strata of household food animals | 0.81 (0.57, 1.16) | 1.15 (0.70, 1.91) | - | - |
| **Distance to nearest CFO drainage path** | >500 m | 1.00 (ref) | 0.88 (0.50, 1.53) | - | - |
| | 101–500 m | 1.10 (0.76, 1.60) | 1.32 (0.71, 2.46) | 0.619 | 0.18 (−0.53, 0.89) |
| | ≤100 m | 0.85 (0.46, 1.57) | 1.80 (0.94, 3.45) | 0.108 | 0.85 (−0.10, 1.81) |
| | 101–500 m within strata of household food animals | 1.10 (0.76, 1.60) | 1.15 (0.73, 1.83) | - | - |
| | ≤100 m within strata of household food animals | 0.85 (0.46, 1.57) | 1.24 (0.82, 1.86) | - | - |

RRs and robust 95% CIs estimated using log-binomial regression models with generalized estimating equations included interaction terms between commercial and household food animal exposure variables and included the following covariates: caregiver education, asset score, child age and sex, and child antibiotic use in the last 3 months. *N* = 1,940 observations across 1,677 child fecal samples (including 910 total 3GCR-EC isolates) for 594 children. CFO = commercial food animal operation. RR: relative risk. CI: confidence interval. 3GCR-EC: third-generation cephalosporin-resistant *E. coli*. RERI: relative excess risk due to interaction.

interaction was near-significant with an interaction *P*-value of 0.108 (RERI: 0.85; 95% CI: −0.10, 1.81) (Table 5). These findings were also robust to sensitivity analyses (Table I in S1 Files). Additional risk factors of ESBL-EC carriage among children in our study site included placing animal feces on household land/crops (RR: 1.63; 95% CI: 1.09, 2.46), household dog ownership (RR: 1.35; 95% CI: 1.00, 1.83), and child pet contact in the last 3 months (RR: 1.54; 95% CI: 1.10, 2.16) (Fig 4 and Table K in S1 Files).

## Discussion

In this study of 7 semirural parishes in Ecuador, we found that increased density of commercial food animal production facilities, household food animal ownership, child pet contact, and rarely/never washing hands after contact with animals were risk factors for 3GCR-EC carriage. The combination of owning household food animals and living within 100 m of a drainage flow path from a commercial food animal operation may have increased the risk of ESBL-EC carriage among children. Other risk factors for ESBL-EC carriage were household dog ownership, child pet contact, and placing animal feces on household land/crops. Clinically relevant STs such as ST 10, ST 131, ST 38, and ST 117 associated with extraintestinal pathogenic *E. coli* (ExPEC) were detected in child fecal samples, highlighting the public health significance of community-acquired ESBL-producing *E. coli* carriage [58]. The results of this study emphasize the need for a One Health approach in the control and prevention of antibiotic-resistant bacterial (ARB) infections, particularly in the context of globally expanding commercial food animal production.

Epidemiologic studies have established a clear link between commercial food animal production and ARB carriage among commercial animal farm workers, their household contacts, and community members [19,59–63]. Epidemiological research on associations between small-scale food animal production and community-acquired resistance in humans, however, has been limited. Previous studies have used samples that are not spatiotemporally matched with household-level exposures, cross-sectional study designs, small sample sizes, or descriptive statistics, only [64–67]. These methodological limitations prevent the reliable estimation of associations between small-scale food animal exposures and community-acquired antibiotic-resistant infections. This study attempted to address these gaps by leveraging repeated measures data from 600 households in which household-level exposure data—including data on ARB carriage in household animals—are spatiotemporally matched with outcome data. With our robust longitudinal One Health study design, we were able to estimate the impacts of food animal exposures, domestic animal exposures, and hygiene practices on antibiotic-resistant *E. coli* carriage in children.

Prevalence estimates in the literature suggest that community-acquired ESBL-EC carriage in healthy populations is increasing globally, with a recent pooled estimate of 21% in 2015 to 2018 [16]. However, there are few studies estimating community-acquired ESBL-producing infections in South America, and estimates are variable. Bezabih and colleagues (2021) estimated a pooled ESBL-EC prevalence of <10% for the Americas, while a 2015 review of ESBL-E estimated a prevalence of 2% for the Americas [68]; both reviews included estimates from studies from the United States in pooled estimates and did not include any studies from Ecuador. Of note, studies in these reviews used selective media to screen for 3GCR-EC or ESBL-EC prior to ESBL confirmatory testing, comparable to the methods used in the present study. A 2008 multicountry study in South and Central America (not including Ecuador) detected ESBL-producing bacteria in 31% of community-acquired intra-abdominal infections [69]. The high prevalence of 3GCR-EC, ESBL-EC, and MDR *E. coli* carriage among healthy children in our study site is concerning. Over 60% of children were carriers of 3GCR-EC with

frequent detection of $bla_{CTX-M}$ genes and 25% were carriers of ESBL-EC at least once throughout the study period, with 86% of all 3GCR-EC being MDR. Even among non-ESBL-producing *E. coli* isolates based on phenotypic testing, *bla* genes encoding for extended-spectrum beta-lactam resistance were detected in over 75% of 3GCR-EC isolates. CTX-M-encoding genes $bla_{CTX-M-55}$, $bla_{CTX-M-65}$, and $bla_{CTX-M-15}$ were the most frequently detected *bla* genes in this food animal-producing region of Ecuador. $bla_{CTX-M-55}$, $bla_{CTX-M-65}$, and $bla_{CTX-M-15}$ are dominant in food animals such as chickens, pigs, and cattle, as well as meat products in China, South Korea, Hong Kong, Canada, Portugal, and elsewhere [70–74]. The distribution of CTX-M-encoding genes in our study site suggests that food animal production plays a critical role in driving the community spread of 3GCR-EC and ESBL-EC in this region of Ecuador.

Other analyses of data collected for this cohort study have previously provided evidence for both horizontal gene transfer and clonal spread of 3GCR-EC in children and animals within and between households in the study site [50,75]. Animal waste management and handling practices were poor in a majority of households with clonal relationships between 3GCR-EC in children and animals [50]. Though evidence of resistant bacteria transmission between backyard chickens, dogs, and humans has been documented in this study site and study period through previous analyses [50,75], it is unclear whether antibiotic use in household food animals and domestic animals or human antibiotic use is driving selection of resistant bacteria in these communities. In fact, reported antibiotic use in children and household animals was low in this study site and participants had limited knowledge about antibiotics and antibiotic stewardship [34]. A recent qualitative study in the same communities found that small-scale poultry and livestock producers typically rely on low-cost traditional veterinary practices rather than administering antibiotics [76]. Notably, antibiotic use in children and domestic animals declined throughout the study period. One possible explanation is that public health measures such as the SARS-CoV-2 pandemic lockdowns may have curbed infectious disease transmission in humans, reducing the need for antibiotic treatment [77]. This hypothesis is supported by the apparent reduction in reported child illness from cycle 3 (before pandemic lockdowns) to cycle 5 (after lockdowns began) in our study. These observed secular trends appear to be nondifferential by household food animal ownership (Table L in S1 Files).

While antibiotic use in household food animals may not be a primary driver of resistance in this area, commercial-scale production operations administering high volumes of antibiotics may be a key driver of emergence and selection of resistant bacteria [1,19]. Results from our study suggest that household food animals and domestic animals still play an important role in determining risk of antibiotic-resistant *E. coli* carriage among community members. Household animals may act as a vector for transmission of resistant bacteria between environmental reservoirs contaminated by commercial food animal waste and humans with whom they come into contact. GPS-tracked movement patterns of free-range poultry in northwestern Ecuador confirmed that backyard chickens (not given antibiotics) travel an average of 17 m from their household and that this range overlapped with small-scale farms of broiler chickens (given antibiotics) [78]. Free-roaming dogs are also common in urban and rural Quito [79] and have been shown to roam up to 28 km from their homes on average in rural Southern Chile [80]. While some backyard chickens in our study site were kept in coops, some free-range chickens and free-roaming dogs may have been exposed to ARB in environmental contamination from nearby commercial operations. For example, animals may be exposed to high levels of fecal contamination in irrigation canals [81], though household animals drinking irrigation water was not identified as a significant risk factor for human ARB carriage in our study. Household members may be exposed to ARB through contact with companion animals, food animals, or their feces. In fact, children with recent contact with pets, from households with ESBL+ animal feces in the yard, or from households that applied animal waste to household food crops had a

slightly greater risk of ESBL-EC colonization in our study. We also reported increasing RRs of ESBL-EC for children in households that both owned food animals and were close to a commercial food animal operation drainage path, which often feed into irrigation canals. Furthermore, the irrigation canals run throughout the study site, which may, in part, drive the geographic spread of beta-lactamase genes we detected in parishes with both high and low numbers of commercial production facilities.

A limitation of the present study is the change in the number of *E. coli* isolated per sample midway through the study. This may impact the probability of detecting the outcome; for example, phenotypic ESBL-EC would be more likely to be detected for a child with multiple isolates tested for ESBL production compared to a child with only 1 isolate tested. We attempted to address this by analyzing data at the isolate level and adjusting for imbalanced data in our statistical approach rather than aggregating the binary outcome at the individual level. We also included a sensitivity analysis using only the first isolate per sample. Despite some slight differences in point estimates, the overall findings were robust to sensitivity analyses, suggesting a low risk of bias in our outcome assessment methods. Another limitation is that we only isolated 3GCR-EC in order to improve detection of ESBL-EC, limiting our findings to this specific type of ARB species. We did not identify other ESBL-producing Enterobacterales, such as *Klebsiella pneumoniae*, which also have clinical importance given the high mortality rates associated with ESBL-producing *K. pneumoniae* infections [82]. Finally, the CLSI guidelines indicate the use of CAZ and CAZ/CLA as well as CTX and CTX/CLA for confirmatory ESBL testing; if a ≥5-mm difference in at least one of these combinations is identified, it is correct to consider the isolate tested as an ESBL-producing isolate. For this study, we evaluated ESBL-production based on the CAZ and CAZ/CLA combination disk diffusion test alone. Therefore, some ESBL-negative isolates reported in the manuscript had the potential to be ESBL-positive if we also had evaluated CTX and CTX/CLA. Based on CLSI guidelines and methods used in this study, all of the ESBL-positive isolates detected and reported in our manuscript are ESBL-positive. The difference between the number of isolates with ESBL genes and isolates with ESBL expression (phenotypically resistant) may be due to the methodology used (only using CAZ and CAZ/CLA and not CTX and CTX/CLA).

Additionally, several risk factors were based on caregiver-reported information and are subject to recall bias. We attempted to reduce bias by asking caregivers to recall weekly frequencies of events (e.g., child contact with animals) over longer periods of time (e.g., 3 months), rather than asking for "yes/no" responses. Finally, this study was halted in the middle of the fourth cycle of data collection due to SARS-CoV-2 lockdowns in March 2020 and did not resume until April 2021, resulting in significant participant dropout. Prior to the pandemic, there was some dropout to do migration out of the study site. This loss to follow-up may have induced some selection bias, though we aimed to address the imbalanced nature of the data in our statistical approach using semiparametric generalized estimating equations with an exchangeable working correlation.

A significant challenge in identifying the source of ARB remains: There is a lack of available data on antibiotic usage and resistance in intensive, commercial food animal operations and their effluent in Ecuador and globally due to limited oversight and surveillance. To accurately characterize the extent to which antibiotic use in commercial production drives community-acquired antibiotic-resistant infections, policies should require that commercial food animal operations monitor and report antibiotic use and conduct routine surveillance for antibiotic-resistant bacteria in food animals, food animal production waste, nearby environmental reservoirs, and food animal products. Future studies should attempt to collect this data to characterize ARB in commercial food animal production settings in Quito. Local and national policies requiring improved waste management practices in large-scale commercial food animal

production operations are also lacking, especially in LMICs [83]. Strategies may include containing and/or diverting animal waste, moving animal grazing areas away from waterways, transporting excess waste to more remote areas with sufficient non-food crop land to apply waste as fertilizer, and monitoring levels of nutrients and antibiotic-resistant bacteria in soil and waterways near discharge points. Governments could implement a permitting process that requires commercial operations to submit a waste management plan for approval by agriculture and environmental protection agencies. For example, in the United States, the Environmental Protection Agency established the National Pollution Discharge Elimination System, which regulates the discharge of pollution from point sources at large animal feeding operations and other industrial sites to bodies of water. Agencies could provide subsidies to support producers with initial costs of improving waste management. Policies and regulations should be tailored to farm size; some practices may be cost-prohibitive for small-scale producers, so appropriate financial incentives should be used to promote best management practices among small food animal farms [84]. Finally, national policies should follow global recommendations to restrict the use of clinically important antibiotics like third-generation cephalosporins in food animals [85]. Restricting antibiotic use in cattle farms has been shown to reduce detection of $bla_{\text{CTX-M}}$ genes in cattle [86]. To improve the effectiveness of such policies, antibiotic stewardship training programs from a One Health lens could be offered to physicians, veterinarians, and food animal producers with an emphasis on the growing antibiotic resistance crisis.

Our study underscores the need for increased monitoring of waste management practices and improved surveillance of antibiotic use and community-acquired antibiotic resistance in LMICs with widespread food animal production. Increased contact with domestic animals, household food animal ownership, and proximity to large-scale food animal production operations—especially in high-density areas—were associated with an increased risk of antibiotic-resistant bacteria carriage in young children. With zoonotic infectious disease risks and hygiene-related prevention strategies at the forefront of public health messaging due to the SARS-CoV-2 pandemic, national governments should prioritize policies and communication strategies that promote improved food animal waste management and safe hygiene practices to reduce the prevalence of ARB carriage and infections.

## Supporting information

**S1 Files. Supplementary files. Checklist:** Strengthening the Reporting of Observational Studies in Epidemiology (STROBE) checklist of items that should be included in reports of cohort studies. **Sample Size and Power Calculations. Table A.** Prevalence of third-generation cephalosporin-resistant, extended-spectrum beta-lactamase, multidrug-resistant, and extensively drug-resistant *E. coli* among children. **Table B.** Proportion of third-generation cephalosporin-resistant *E. coli* (3GCR-EC) isolates resistant to individual antibiotics in phenotypic susceptibility testing by data collection cycle. **Table C.** Antibiotic resistance of 3GCR-EC isolates from animal fecal samples (1 colony isolated per fecal sample) collected at the same households as child fecal samples, stratified by animal species. **Table D.** Prevalence of clinically important sequence types (ST) among sequenced 3GCR-EC isolates (*N* = 571) from child fecal samples. **Table E.** Proportion of 3GCR-EC isolates with beta-lactamase genes (among 15 most prevalent) detected in whole-genome sequences, stratified by phenotypic ESBL production. **Table F.** Prevalence of beta-lactamase resistance genes among sequenced 3GCR-EC isolates from children, stratified by parishes and intensity of commercial food animal production. **Table G.** Average number of total antibiotic resistance genes (ARGs) per third-generation cephalosporin-resistant *E. coli* isolate from children, stratified by parish. **Table H.** Prevalence

of CTX-M-type genes among sequenced 3GCR-EC isolates from children, stratified by parish and intensity of commercial food animal production. **Table I.** Sensitivity analysis results for main analysis associations between combined food animal exposures and 3GCR-EC and ESBL-EC including only 1 isolate per child fecal sample. **Table J.** Associations between secondary risk factors and 3GCR-EC carriage among children. **Table K.** Associations between secondary risk factors and ESBL-EC carriage among children. **Table L.** Secular trends in caregiver-reported child illness and antibiotic use stratified by household food animal ownership. **Table M.** Access to water and sanitation at households included in the main analysis ($N$ = 594). **Fig A.** Directed acyclic graph of causal relationship between exposures to commercial and household food animal production and ESBL-*E. coli* carriage in children. SES: socioeconomic status. ESBL: extended-spectrum beta-lactamase *E. coli*. **Fig B.** Flow chart of enrollment and follow-up by data collection cycle. Households were included in the final analysis if they had the necessary exposure, outcome, and covariate data. **Fig C.** Prevalence of beta-lactamase genes (top 15 most prevalent) among sequenced third-generation cephalosporin-resistant *E. coli* isolates from children, stratified by phenotypic extended-spectrum beta-lactamase (ESBL) production. **Fig D.** Prevalence of beta-lactamase genes by type among third-generation cephalosporin-resistant *E. coli* (3GCR-EC) isolated from children, stratified by parish. (DOCX)

## Acknowledgments

We sincerely thank the data collection team and community partners in Ecuador for their hard work, time, and dedication to this research, especially through the SARS-CoV-2 pandemic. Thanks to Kathleen Kurowski and Rachel Marusinec for their contributions to laboratory work during the initial year of this study. Thanks to Professor Lisa Barcellos, Professor Ayesha Mahmud, and Professor Ellen Eisen for their invaluable input on the methodological approach in the early phases of this analysis. We are deeply saddened by the loss of our co-author, Dr. Lee Riley, who was a kind mentor, outstanding scientist, physician, researcher, and good friend. Finally, we are extremely grateful for the generosity and commitment of our study participants in Quito, without whom this work would not be possible.

## Author Contributions

**Conceptualization:** Heather K. Amato, Gabriel Trueba, Jay P. Graham.

**Data curation:** Heather K. Amato, Fernanda Loayza, Liseth Salinas, Diana Paredes, Daniela Garcia, Timothy J. Johnson.

**Formal analysis:** Heather K. Amato.

**Funding acquisition:** Gabriel Trueba, Jay P. Graham.

**Investigation:** Fernanda Loayza, Liseth Salinas, Diana Paredes, Daniela Garcia, Soledad Sarzosa, Carlos Saraiva-Garcia, Timothy J. Johnson, Gabriel Trueba, Jay P. Graham.

**Methodology:** Heather K. Amato, Amy J. Pickering, Lee W. Riley, Gabriel Trueba, Jay P. Graham.

**Project administration:** Fernanda Loayza, Soledad Sarzosa, Carlos Saraiva-Garcia.

**Resources:** Timothy J. Johnson, Gabriel Trueba, Jay P. Graham.

**Supervision:** Amy J. Pickering, Lee W. Riley, Gabriel Trueba, Jay P. Graham.

**Validation:** Heather K. Amato.

**Visualization:** Heather K. Amato.

**Writing – original draft:** Heather K. Amato.

**Writing – review & editing:** Heather K. Amato, Fernanda Loayza, Liseth Salinas, Diana Paredes, Daniela Garcia, Soledad Sarzosa, Carlos Saraiva-Garcia, Timothy J. Johnson, Amy J. Pickering, Lee W. Riley, Gabriel Trueba, Jay P. Graham.

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
