## [Editor Report · Decision Letter 0]

22 Aug 2022

Dear Dr Amato, 

Thank you for submitting your manuscript entitled "Risk factors for extended-spectrum beta-lactamase (ESBL) producing E. coli carriage among children in a food animal producing region of Quito, Ecuador" for consideration by PLOS Medicine.

Your manuscript has now been evaluated by the PLOS Medicine editorial staff as well as by an academic editor with relevant expertise and I am writing to let you know that we would like to send your submission out for external peer review.

Please re-submit your manuscript within two working days, i.e. by Aug 24 2022 11:59PM.

Kind regards,

Philippa

Philippa Dodd, MBBS MRCP PhD

Senior Editor

PLOS Medicine

---

## [Decision Letter · Decision Letter 1]

7 Dec 2022

Dear Dr. Amato,

Thank you very much for submitting your manuscript "Risk factors for extended-spectrum beta-lactamase (ESBL) producing E. coli carriage among children in a food animal producing region of Quito, Ecuador" (PMEDICINE-D-22-02768R1) for consideration at PLOS Medicine. 

[LINK]

In light of these reviews, I am afraid that we will not be able to accept the manuscript for publication in the journal in its current form, but we would like to consider a revised version that addresses the reviewers' and editors' comments. Obviously we cannot make any decision about publication until we have seen the revised manuscript and your response, and we plan to seek re-review by one or more of the reviewers. 

We expect to receive your revised manuscript by Dec 28 2022 11:59PM. Please email us (plosmedicine@plos.org) if you have any questions or concerns.

We look forward to receiving your revised manuscript. 

Sincerely,

Philippa Dodd, MBBS MRCP PhD

PLOS Medicine

plosmedicine.org

GENERAL

Please respond to all editor and reviewer comments below, in full.

Please number each line of the manuscript starting with 1 on the first line of page 2 and in continuous sequence thereafter.

Please ensure that the study is reported according to the STROBE guideline, and include the completed STROBE checklist as Supporting Information. Please add the following statement, or similar, to the Methods: "This study is reported as per the Strengthening the Reporting of Observational Studies in Epidemiology (STROBE) guideline (S1 Checklist)."

ABSTRACT 

Abstract methods and findings:

Please include the actual amounts and/or absolute risk(s) of relevant outcomes (including NNT or NNH where appropriate), not just relative risks or correlation coefficients. (example for absolute risks: PMID: 28399126).

Where 95% CIs are reported please also report p-values. 

Suggest for clarity that statistical information is defined clearly perhaps with abbreviations (e.g. 95% CI) at each reporting, as you have done in the main manuscript, as opposed to just once in full.

Please ensure that all numbers presented in the abstract are present and identical to numbers presented in the main manuscript text.

It may be helpful to elaborate on the details of the study design including for clarity some of the different risk factors (or categories of risk factor) investigated. The reader doesn’t learn of any until the significant results are reported. Suggest including the number of participants (as well as the sample number), length of follow-up/number of visits and more clearly defining the region (see also statistical reviewer comments) and use of animals for subsistence farming Vs trade

Suggest moving the following sentence “We used multivariable log-binomial regression models to estimate relative risks (RR) of 3GCR-EC and ESBL-EC carriage.” To follow the description of the study before the results are presented.

Please include the important dependent variables that are adjusted for in the analyses.

In the last sentence of the Abstract Methods and Findings section, please describe the main limitation(s) of the study's methodology

Abstract Conclusions:

Please address the study implications without overreaching what can be concluded from the data; the phrase "In this study, we observed ..." may be useful.

Please interpret the study based on the results presented in the abstract, emphasizing what is new without overstating your conclusions.

Please avoid vague statements such as "these results have major implications for policy/clinical care". Mention only specific implications substantiated by the results.

Please avoid assertions of primacy ("We report for the first time....")

AUTHOR SUMMARY

METHODS and RESULTS

Did your study have a prospective protocol or analysis plan? Please state this (either way) early in the Methods section.

For all observational studies we ask that the following is clearly indicated in the manuscript text: 

(1) the specific hypotheses you intended to test, 

(2) the analytical methods by which you planned to test them, 

(3) the analyses you actually performed, and 

(4) when reported analyses differ from those that were planned, transparent explanations for differences that affect the reliability of the study's results. If a reported analysis was performed based on an interesting but unanticipated pattern in the data, please be clear that the analysis was data-driven.

Page 11 para 3: please ensure that where you report percentages that the numerators and denominators used to derive them are clearly reported. In the paragraphs above whole numbers are reported but it is a bit arduous to the reader to go back and forth for reference.

FIGURES 

Please consider avoiding the use of red or green to make the figures more accessible to those with colour blindness

Figure 1: Please confirm that the appropriate usage rights apply to the use of this map. Please see our guidelines for map images: 

https://journals.plos.org/plosmedicine/s/figures#loc-maps

Figure 2: PLOS medicine usually advises against the use of pie charts but in this specific case we agree with the methodological reviewer that they serve their purpose here very nicely

Figures 3 and 4: Please clearly indicate the meaning of lines and dots on the plots. Please clearly indicate whether the analyses are adjusted and in the figure captions what factors they are adjusted for. Where adjusted analyses are presented, please also present unadjusted analyses. It would be helpful to the reader to list the numerical values for RR and 95% CIs in a column next to the plots. In addition, where you report 95% CIs please also report p-values and in the figure caption please define the test used to derive them

Please ensure all abbreviations for statistical reporting are defined e.g RR and CI

TABLES

For the main outcome measures presented in table 3 and 4, to help facilitate transparency of data reporting please indicate in the column headers that these analyses are adjusted and also provide the unadjusted analyses for comparison

SUPPLEMENTARY FIGURES AND TABLES

Please check carefully throughout and ensure all abbreviations are defined (for example SES, ESBL).

Where you report 95% CIs please also ensure p-values are reported

Where adjusted analyses are reported please clearly indicate which factors are adjusted for, in the caption, and also present the unadjusted analyses for comparison.

DISCUSSION

Please present and organize the Discussion as follows: a short, clear summary of the article's findings; what the study adds to existing research and where and why the results may differ from previous research; strengths and limitations of the study; implications and next steps for research, clinical practice, and/or public policy; one-paragraph conclusion.

REFERENCES

Please use the "Vancouver" style for reference formatting, and see our website for other reference guidelines 

https://journals.plos.org/plosmedicine/s/submission-guidelines#loc-references

Citations should be in square brackets, and preceding punctuation [1,3,5,7] or [1-4,7]. Please note the absence of spaces between citations.

Journal name abbreviations should be those found in the National Center for Biotechnology Information (NCBI) databases. 

In the bibliography, please list up to but no more than 6 names followed by et al where more than 6 authors contribute to a study.

PARACHUTE RESEARCH

We note that you conducted research or obtained samples in a foreign country. Did you consider including a local author as first or last author? If not, we recommend that you consider doing so in line with ICMJE's authorship requirements (https://www.icmje.org/recommendations/browse/roles-and-responsibilities/defining-the-role-of-authors-and-contributors.html). PLOS has a parachute research policy which aims to promote collaboration and inclusivity in global health research. You are required to complete PLOS’ questionnaire on inclusivity in global research and submit it with your revised paper. The policy and questionnaire can be found at https://journals.plos.org/plosone/s/best-practices-in-research-reporting.

DECLARATION STATEMENTS

Please remove these from the end of the manuscript and place in the manuscript submission form only. The ethics statement can be placed in the methods section.

SOCIAL MEDIA

In the event that your paper is published, to help us extend the reach of your research, please provide any Twitter handle(s) that would be appropriate to tag, including your own, your coauthors’, your institution, funder, or lab. Please respond to this email with any handles you wish to be included when we tweet this paper.

Comments from the reviewers:

Reviewer #1: This manuscript describes the results of an observational study on environmental exposure by commercial and small-scale food animal production and risk factors for ESBL-producing E. coli and third-generation cephalosporin-resistant E. coli carriage in children in Ecuador. It is important that this kind of studies are performed in low- and middle-income countries to increase the knowledge and to identify potential measures that can be taken to reduce the risks for ESBL carriage in children. Furthermore, it is a nicely designed and well-written study with repeated measurements per household. I do have some questions for the authors though.

Major comments:

1. Why were the selected E. coli isolates frozen and thawed again before susceptibility and phenotypic ESBL-production testing was performed? Do the authors think this may have influenced the results they found compared to immediate susceptibility and phenotypic ESBL-production testing?

2. Why were the analyses performed on isolate level, and not on child-level (child being colonized or not)? Did the authors check for results on child-level? I am also curious on parish level how many kids were colonized, and the mean/sd number of isolates on child-level per parish. 

3. Hoe were the isolates selected that were sequenced? It is not mentioned in the Methods section how this selection was done.

4. Page 11 second paragraph: 1,677 child fecal samples from 594 households remained. Is it then correct that the 1,677 fecal samples were taken from 594 children? Also, I don't get how the numbers mentioned in this paragraph add up to 1,940 observations: is that 904 3GCR-EC isolated in children plus 1,060 3GCR-EC in animals (=1,964)? Please clarify.

5. A lot of different statistical analyses and tests were performed. However, I did not read anything about corrections made for multiple testing which I think is very appropriate in this case. 

6. Table 1: Is it correct that none of the household food animals received antibiotics in the 5th data collection cycle? In the other cycles it ranged from 9.5-26.4%, so this is a big difference. Do the authors have an explanation for this finding?

7. Longitudinal findings in individual children, did the authors analyse whether the same children carried the same ST E. coli and gene during follow-up data collection cycles? Of course, this is only possible for children who participated >1 cycle. This would also be very interesting to see, whether there was a new or new type of colonization or whether it is still the same type. In the last case, it is could be a non-cleared colonization or a repeated exposure/colonization.

8. Table S6: high percentage of OXA genes in Tababela compared to all other parishes, explanation for this difference.

9. Figure 2/Table S8: most of the commonly detected CTX-M-type genes are both found in parishes with low- and high-CFOs. For example, CTX-M-15 represents 50% of the CTX-M genes found in Tababela with low CFO and ≈55% in El Quinche with high CFO. Are there any other suspected common sources that the authors could think of which may explain these findings?

10. Results page 14/15 and 1st paragraph of the Discussion. All results with an RR >1 are presented as having an increased risk, but some of them are not statistically significant since the 1 is included in the 95% CI. In my opinion this should be mentioned more explicitly when results are or are not statistically significant. For example: "However, the combination of owning household food animals and living <1.5 km from the nearest commercial operation was associated with an increased risk in 3GCR-EC carriage (RR: 1.15; 95% CI: 0.97, 1.33) compared to those without food animals who lived further from commercial food animal operations". It is presented like a significant increased risk, but it is not. Also, the last line of page 14: "chicken ownership (RR: 1.10; 95% CI: 0.98, 1.24), and rarely/never washing hands after contact with animals (RR: 1.15; 95% CI: 0.98, 1.34)". These are non-significant risk factors while the first two factors mentioned in this sentence are not. I would not mention them separately as these non-significant factors can be found in the Tables. Finally, this non-significant "rarely/never washing hands after contact with animals" is even mentioned in the abstract together with significant results.

11. Table 4: The RR of in children without household food animals with distance to the nearest farm <1.5km away seems to be protective (although not statistically significant) compared to >=1.5km away. Any explanation for that?

12. Page 17 line 2: Were animal samples and child samples matched? So were the same ST's and genes identified in children and animals of the same household? And were analyses performed on these matched isolates?

Minor comments:

1. Page 3 last paragraph: "ESBL-producing enterobacteriaceae infections". "Enterobacteriaceae" are renamed as "Enterobacterales".

2. Table S3: The column name of the second column is 'H', I think this reflects the number of households. The legend says it should read 'HH'. Please adapt them consistent.

3. Table S10: I suggest to present the RR and 95% CI with 2 decimals (like in all other Tables) and the P-values with 3 decimals instead of 4.

Reviewer #2: Dear editor,

Thank you for asking me to review this article. I believe the topic fits Plos Medicine really well. Livestock is already known to be a reservoir for antimicrobial resistant bacteria and livestock-human transmission is documented as well. However, most of the studies are conducted in the Western world. I believe it is very interesting to investigate the risk of animal food production for ESBL carriage in humans in other parts of the world. Therefore, I do believe that after improvement this manuscript can add valuable information to the topic of human antimicrobial carriage. However, I have some major concerns regarding the data analysis performed, interpretation of the results and following conclusions stated. Therefore, I would advise major revisions.

I would recommend that the authors check the data analysis and thoroughly read through their manuscript again and rewrite parts of it and also reduce the length of the manuscript. In the section "Comments to the authors" I also commented in more detail regarding abovementioned aspects.

Comments to the authors

I agree with the authors that it is very important to investigate livestock as a risk factor for human carriage of antimicrobial resistant bacteria (or genes) in low and middle income countries, therefore I stress the importance of the topic. However, I have some concerns regarding certain aspects (mainly your data analysis and discussion) of the manuscript. Moreover, I believe you can shorten the manuscript. Below you will find some comments in more detail.

Abstract

Page 2, Conclusions: although I understand the reasoning of what is stated in these conclusions, I don't think this conclusion aligns well with the study conducted. Risk factors for carriage were under investigation, not policies and interventions to curb the spread. 

Introduction

I find the introduction too long. Every paragraph should be shortened. I think the order of paragraphs can be improved

First paragraph, first sentence: I don't understand what you state. Contamination with what? You might consider deleting this first sentence.

First paragraph, ref Seifert et al. 2013: There is more recent work done that you can cite. I would also use more references for such a statement.

First paragraph, last sentence: Please specify which communities. Moreover, although livestock-human transmission is important, but I would mention human-human transmission as background information as well. 

Second paragraph: this part described the urgency and importance. I would place this paragraph first. This is the reason why you want to investigate risk factors for this human carriage. 

Third and fourth paragraph can be combined. 

Methods

The first paragraph could use the subheading 'study design'. I think the section on data analyses is hard to follow. I would suggest the authors to be more clear and precise. A figure might help to visualize it.

Page 5: What is the reasoning for the inclusion criteria 'child between 6 months and 5 years'?

Page 6, 'Density…..(Amato et al. 2020)': This is background information/interpretation and should not be discussed in method section, but in introduction and/or discussion.

Page 7, outcome assessment: What do you mean by animal that were present at the household? Only domestic animals? Or food production animals (backyard farming) as well?

Page 9, statistical analyses: How did you account for potential interaction? What do you mean by three different models? 

What do you mean by 'using cut-points ….. policy relevance'?

How where the results from the animal derived isolates used in the models?

What do you mean by assessed at the isolate level? Did you model at the isolate level? I would say it makes sense to model at the child level (= sample level). 

Results

I think the results section can be shorter in general. Lot of percentage and gene types are described, which can easily be seen in the Table. I would focus on highlights in text. 

Page 10/11: I find the figure S2 helpful in understanding. However, from both the text and the figure I cannot interpret how many households did how many repeats? Please provide this information?

Page 11: What are the 1940 observations? It looks like you are modeling at the isolate level. But exposures are at the child sampling moment level (i.e. each stool sample is a single observation). 

Page 12, Food Animal Production & Domestic Animals, first paragraph: what is the rationale for only mentioning the first four cycles of data collection to describe the distance to the number of farms?

Page 14, Risk Factors for 3GCR-EC: I am slightly confused by these results. You state that children with > 5 commercial food animal operations in a 5-km radius of their household had a higher risk of 3GCR-EC carriage. However, in table 3 I only see the results of the stratified analyses. Without stratifying, what is the effect? In order to understand any effect modification, I would like to see the overall results as well. Why is a different cut-of chosen for the no of operation in an 5 km radius in the analysis among those with household food animals. Also, I am missing the sizes of the strata.

Discussion

I think most of the paragraphs in the discussion can be shortened, it often takes a long read before you reach your message. The paragraph on policy advice should be deleted in my opinion. This was not your research question in the first place, and I think your results should be used for policy if applicable, but policy is not supposed to be made by researchers.

Page 16, second paragraph: I do not think that the epidemiological studies have shown a clear link between food animal production and community-acquired antibiotic-resistant infections. Instead it is associated with carriage of ABR bacteria. 

Page 17, top of page: I agree that repeated measurements are essential to investigate this association, but from the manuscript the range of number of repeated measurements is not clear.

Also, you mention your One Health design, what do you mean by that? I read that you samples animals (domestic or commercial as well?) at the same household, but I don't see how you used it in your analysis. 

Page 17, paragraph on AB use: I find this paragraph hard to follow. Please be clear on what is from the other study and what is from your study.

What do you mean by indirect curbing of infectious diseases in animals by the pandemic? 

Page 18, bottom of the page: I don't agree with your analysis at the isolate level (see comments on methods and results)

Page 19, paragraph on prevalence. I would move this paragraph up, since it is a rather important finding. 

Page 20, gene types found in children: I understand that the gene types found are similar as in livestock animals. What are the main types in the general population lesser in contact with animals? That should give a more or less control situation.

Page 20: delete the paragraph on policy and please remove the policy regarding statements also from your conclusions and abstract. 

Reviewer #3: This is a potentially interesting study on a relevant topic. However, it is presented as an epidemiological study, which highlights a number of shortcomings in the design and reporting in the paper. The study mostly uses proxy indicators of AMR exposure, or caregiver reported indicators of hygiene and child behaviour. Any observed associations do not really shed light on the pathways of AMR transmission. 

The key limitation is the lack of assessment of antibiotic resistant E. coli in the majority of the exposures that are reported.

Specific comments:

Assessment of risk factors is done using proxy measures (e.g. geographical distance and concentration of commercial livestock/animal farms) without any direct measures of antibiotic resistant bacteria, genes or antibiotic residues and there is no information presented on whether these commercial premises used antibiotics, which antibiotics were used, and/or gut colonisation with ARB in the commercial animals. 

Similarly, the proxy measure of proximity to drainage channels does not provide direct evidence of exposure to ARB/ARG. No analysis of water samples from these drainage channels were taken or analysed.

Proxy measures can only provide weak inferences about relationships (or not) between environmental exposures and human colonisation with ARB.

'The primary outcomes, 3GCR-EC and ESBL-EC carriage, were assessed at the isolate level' There is no clear rationale for why the study reports AMR at the isolate level rather than the individual child (or household animal) level, especially as the methods changed part way through the study (extracting 5 E. coli isolates in the early rounds, and only 1 isolate in later rounds). 

As this is intended as an epidemiological study, why is the analysis not at the individual child level, with the denominator as the total number of individuals, taking into account repeat measures?

If the study places importance on caregiver-reported hygiene and reported infant behaviours (child handwashing, contact with animals or pets) as risk factors for child colonisation then some validity of these measures is needed. These types of indicators have poor validity & reliability in most contexts. A reporting window of child pet contact in the last 3 months (yes or no) is also a very crude indicator.

The key limitation is the lack of assessment of antibiotic resistant E. coli in the majority of the exposures that are reported. 

Other suggestions/queries.

The authors make it clear that the number of isolates per sample changed part way. However it is not clear why the analysis did not use one isolate per sample throughout all the analysis to remove the bias, then there would not be the need for a sensitivity analysis. 

Some of the key information for the study seems to be in supplementary tables e.g. point prevalence of child ESBL-EC colonisation at each survey, ranging from 7% to 16%, average 11%, which seems relatively low. This would merit reporting in the results and discussion.

Samples were left with participants/households for 24 hours, how was it ensured that these were actually stored in fridges or on ice?

Reviewer #4: See attachment

Michael Dewey

[LINK]

---

## [Decision Letter · Decision Letter 2]

15 Jun 2023

Dear Dr. Amato,

Thank you very much for re-submitting your manuscript "Risk factors for extended-spectrum beta-lactamase (ESBL) producing E. coli carriage among children in a food animal producing region of Ecuador" (PMEDICINE-D-22-02768R2) for review by PLOS Medicine.

I have discussed the paper with my colleagues and it was also seen again by 3 reviewers. I am pleased to say that provided the remaining editorial and production issues are dealt with we are planning to accept the paper for publication in the journal.

[LINK]

We look forward to receiving the revised manuscript by Jun 22 2023 11:59PM.   

Sincerely,

Philippa Dodd, MBBS MRCP PhD

PLOS Medicine

plosmedicine.org

Requests from Editors:

GENERAL

Please respond to all editor and reviewer comments detailed below, in full.

TITLE

Please revise your title according to PLOS Medicine's style. Your title must be nondeclarative and not a question. It should begin with main concept if possible. "Effect of" should be used only if causality can be inferred, i.e., for an RCT. Please place the study design ("A randomized controlled trial," "A retrospective study," "A modelling study," etc.) in the subtitle (ie, after a colon). Please include the country of origin (Ecuador) in the title.

AUTHOR SUMMARY

Thank you for including an author summary which reads very nicely but is missing some detail. Please see the information under each of the headings below and revise accordingly (specifically, sample sizes and limitations are notably absent in the current version). Please keep in mind that the summary should consist of 2-3 succinct bullet points under each heading:

• Why Was This Study Done? Authors should reflect on what was known about the topic before the research was published and why the research was needed.

• What Did the Researchers Do and Find? Authors should briefly describe the study design that was used and the study’s major findings. Do include the headline numbers from the study, such as the sample size and key findings. 

• What Do These Findings Mean? Authors should reflect on the new knowledge generated by the research and the implications for practice, research, policy, or public health. Authors should also consider how the interpretation of the study’s findings may be affected by the study limitations. In the final bullet point of ‘What Do These Findings Mean?’, please describe the main limitations of the study in non-technical language.

INTRODUCTION

Suggest moving lines 110-113 ending ‘…mortality [7–10].’ to paragraph 1, line 98 preceding, ‘Anti-biotic resistant bacteria’

Suggest new paragraph at line 104 sentence beginning, ‘With the rapid…’ and combing with sentence at line 113 beginning asymptomatic carriage.

Line 126 – please use one or the other of the two in-text reference callouts.

Line 134 – please change ‘feces’ to ‘faeces’ 

METHODS and RESULTS

Please also see statistical reviewer comments (reviewer #4) below.

In reference to reviewer #4 comments (please see below) please clarify how the sample size was determined.

Line 170 – please state if consent was written or oral

DISCUSSION

Please begin the introduction with a short clear summary of the article’s findings.

Line 512 - please revise to read SARS-CoV-2 also at line 560, please check and amend throughout.

SUPPORTING INFORMATION

Figure S1 – please see statistical reviewer (#4) comments below.

REFERENCES

Throughout, for in-text reference callouts, please remove spaces from between citations. For example, line 100 should read ‘[2,3]’ as opposed to ‘[2, 3]’.

SOCIAL MEDIA

If not already done so, to help us extend the reach of your research, please detail any Twitter handles you wish to be included when we tweet this paper (including your own, your coauthors’, your institution, funder, or lab) in the manuscript submission form when you re-submit the manuscript.

Comments from Reviewers:

Reviewer #1: The revised manuscript has improved considerably and all the points I raised have been answered appropriately. 

A few additional very minor suggestions from my side:

- Line 139: I would not only say that risks of "antibiotic-resistant and ESBL-E infections" are urgently needed to be quantified, but also include carriage here, since this is what the main objective of the study and this data is also still sparce from LMICs. Adapt "antibiotic-resistant and ESBL-E infections" to "antibiotic-resistant and ESBL-E carriage and infections". I would also consider to include this to line 459 and the conclusion (line 604).

- Line 126: reference 18 is included twice.

- Line 439: change "0.1080" into "0.108".

- Line 554: "enterobacteriales" should read "Enterobacterales".

Reviewer #2: I think the manuscript has improved in terms of length and clarity. However, some of my bigger concerns remain partially. Below, you will find my comments in more detail. I would like to stress that I don't understand that the authors decided to stick with the analysis on isolate level instead of analyzing on the individual level (as raised by almost all reviewers).

Introduction

I think the introduction has improved in its focus and it is more concise. However, I think the line in the first two paragraphs is still somewhat shaky. I don't understand some of the choices made in the order of the addressed background information. It switches a few times from ABR to ESBL and back. The same goes for health impact (addressed in both paragraphs). All information is relevant, but can be improved in ordering. 

Materials and Methods

In the inclusion criteria you mention the consent of the caregiver as the third criteria. I would say it is a given that you need consent. I would just make a statement expressing that consent was given by the caretakers for al study participants (or any other covering phrase).

In the statistical analyses it is not stated at what level the outcome is (isolate level, which I still disagree on).

You described a sensitivity analysis with only the first isolate. I would say the same should have been done for the prevalence of carriage (since the sensitivity for finding a positive results is higher when taking more parallel observations per individual). Or at least discuss the impact of a different number of isolates between individuals. 

Results

This part has changed massively. However, I still am confused sometimes about the numbers. In line 332-334 is stated how many isolates were collected from how many children from how many households. By including the number of isolates where cephalosporin resistant it get's less clear. I would not put in outcome results mixed with output results. 

Line 377 and further: In how many of the children with at least one positive isolates, positive isolates were also found in animals? 

Line 388: how were the 200 additional isolates selected?

Line 406: please delete 'all fecal samples', because you are modeling at the isolate level. 

Line 419: do you mean marginally significant? Instead of marginally significant I would suggest to use the term borderline significant. 

If I understood correctly, another manuscript is discussing the similarity between the isolates collected within epidemiologically linked clusters (humans and animals from the same household). The combination of an epi analysis and the isolate characteristics are telling a much fuller view on the matter. I really don't understand why this information is not used in this manuscript to support the findings. 

Discussion

Line 445-449: Please delete, you don't have to summarize the study.

Line 461: Depending on the ARB/ARG, but most of the studies do not report transmission from humans with intense contact with animals to other humans. You might want to nuance this statement a bit.

469-474: Please delete, I don't see any reason to state this. It does not interpret results or highlight any finding in a broader context.

Line 543: I don't agree with your reasoning regarding your choice on what level to perform the analysis. Especially in terms of translating and interpreting the results. A risk factor for a positive isolate is less interesting, since the exposure is present (and therefore also focus for mitigation) on the individual level. And not on the isolate level. I would strongly suggest (as the other reviewers did as well) to model at the individual level and do a sensitivity level as you did with an individual outcome only based on the first isolate. Furthermore, I find the term 'some slight differences' rather vague. 

Line 546: Please replace 1 by one.

Line 566-594: Although I understand the request for including implications, I find this way too long. I am sure you can describe the points you are raising in a much more concise way. 

Reviewer #4: The authors have addressed most of my points but there remain a few to clear up.

I asked whether there had been a formal sample size determination but the rebuttal does not seem to mention this either way.

The authors have estimated parameters in their model page 18 "Multivariable log-binomial regression models were used to estimate unadjusted and adjusted

relative risks" so I still fail to see how they can continue to claim they used non-parametric methods and hence how these were supposed to deal with selection effects.

I am afraid I do not understand their response about Supplementary Figure S1. Either a risk factor is theoretically important in which case it belongs in S1 and in the analysis even if it fails to reach some arbitrary level of statistical significance or it is not in which case it should not appear anywhere. The authors have put household members use of antibiotics in S1 but not the analysis which is inconsistent.

[LINK]

---

## [Decision Letter · Decision Letter 3]

28 Jul 2023

Dear Dr. Amato,

Thank you very much for re-submitting your manuscript "Risk factors for extended-spectrum beta-lactamase (ESBL) producing E. coli carriage among children in a food animal producing region of Ecuador: A repeated-measures observational study" (PMEDICINE-D-22-02768R3) for review by PLOS Medicine.

I have discussed the paper with my colleagues and the guest editor and it was also seen again by the statistical reviewer. I am pleased to say that provided the remaining editorial and production issues are dealt with we are planning to accept the paper for publication in the journal.

[LINK]

We look forward to receiving the revised manuscript by Aug 04 2023 11:59PM.   

Sincerely,

Philippa Dodd, MBBS MRCP PhD

PLOS Medicine

plosmedicine.org

Requests from the Guest Editors:

This study was very laborious it is very good but it has microbiological flaws. I am concerned with the microbiologic aspects of the study, especially items 4 and 5, the authors must explain these before having the manuscript accepted. They are:

1. The identification of the E. coli isolates at the species level was not confirmed by a proper methodology. 

2. The authors used an outdated version of CLSI (2014) for interpreting the antimicrobial susceptibility results. There is an updated version freely available. 

3. The methodology employed for confirming the ESBL phenotype. They use only ceftazidime disks combined with clavulanic acid. 

4. The disproportion between the number of isolates resistant to third-generation cephalosporins and the number of isolates detected as ESBL producers. In carbapenem-susceptible isolates, the production of ESBL and/or AmpC are the main mechanisms of resistance to third-generation cephalosporins in Enterobacterales. Due to the methodology employed, many ESBL-producing isolates could have been missed. It may be true because ESBL encoding genes were detected in "ESBL negative isolates" as explained below.

5. The detection of ESBL encoding genes such as the blaCTX-M genes among isolates classified as non-ESBL producers (ESBL-negative; Table S5 and Figure S3) is a cause of concern. How do the authors justify the presence of the ESBL encoding gene and the absence of the ESBL phenotype? Were the E. coli isolates not detected as ESBL-positives because there was an association with other resistance mechanisms? Was there a lack of gene expression? The gene may not be expressed in some isolates, but it would be uncommon to a high proportion of isolates unless there is a spread of a specific clone. Could this result be justified by the methodology employed to confirm the ESBL phenotype? The authors employed ceftazidime discs combined with clavulanic acid. It would not be a problem for screening some variants of CTX-M, such as CTX-M-3 or CTX-M-15, but it would be for other variants because cefotaxime and ceftriaxone are the preferred substrates for other variants. The authors must clarify the reasons for detecting ESBL-encoding genes in ESBL-negative isolates. Do these isolates express or not these genes? If these genes are expressed, these E. coli isolates must be reclassified as ESBL-positives. The numbers would change and new analyses would be necessary.

6. The authors observed colonization of children (Table S2 - N=5) and animals (Table S3 - N=2) by E. coli resistant to imipenem. The authors did not make any comments on this. Were they really E. coli? Was this resistance phenotype confirmed? If it was, were these isolates producers of carbapenemases? It would be interesting to know if any children’s family members had been recently hospitalized.

Comments from Reviewers:

Reviewer #4: The authors have addressed my few remaining points.

Michael Dewey

[LINK]

---

## [Editor Report · Decision Letter 4]

15 Sep 2023

Dear Dr Amato, 

On behalf of my colleagues I am pleased to inform you that we have agreed to publish your manuscript "Risk factors for extended-spectrum beta-lactamase (ESBL) producing E. coli carriage among children in a food animal producing region of Ecuador: A repeated-measures observational study" (PMEDICINE-D-22-02768R4) in PLOS Medicine.

The Special Issue Guest Editor(s) raise on-going concern regarding the criteria used to classify ESBL isolates, please see the comments below. Prior to publication we require that you include a paragraph in the limitations section of your discussion detailing this as a significant limitation of your study. Please also detail this limitation in the abstract at the end of the methods and findings section and as a final point in the ‘what do these findings mean section of the author summary. We cannot proceed to publication without these additional details.

COMMENTS FROM THE GUEST EDITOR(S)

"The main issue with this manuscript is that the authors used a phenotypic test as a standard to classify the isolates as ESBL and identify the risk factors for ESBL. Even if they had tested cefotaxime disks with clavulanic acid, they might not have identified E. coli isolates that were ESBL producers associated with other resistance mechanisms, such as the production of AmpC (blaCMY-2), as they did, for example.

There is no widely accepted standard, but in my opinion, they should have considered detecting ESBL-encoding genes in third-generation cephalosporin-resistant E. coli isolates as the criterion for classifying isolates as ESBL producers and then identifying risk factors for ESBL colonization from there.

The study is wonderful, involving the efforts and work of many people, with a very thorough analysis of the results. However, it made a mistake in defining the criteria for classifying isolates as ESBL."

PRESS

Best wishes, 

Philippa Dodd, MBBS MRCP PhD 

PLOS Medicine